# Variational Inference with Gaussian Mixture by Entropy Approximation

## Abstract

Variational inference is a technique for approximating intractable posterior distributions in order to quantify the uncertainty of machine learning. Although the unimodal Gaussian distribution is usually chosen as a variational family, it hardly approximates the multimodality. In this paper, we employ the Gaussian mixture distribution as a variational family. A main difficulty of variational inference with the Gaussian mixture is how to approximate the entropy of the Gaussian mixture. We approximate the entropy of the Gaussian mixture as the sum of the entropy of the unimodal Gaussian, which can be analytically calculated. In addition, we theoretically analyze the approximation error between the true entropy and approximated one in order to reveal when our approximation works well. Specifically, the approximation error is controlled by the ratios of the distances between the means to the sum of the variances of the Gaussian mixture. Furthermore, it converges to zero when the ratios go to infinity. This situation seems to be more likely to occur in higher dimensional parametric spaces because of the curse of dimensionality. Therefore, our result guarantees that our approximation works well, for example, in neural networks that assume a large number of weights.

**Keywords:** Gaussian Mixture, Entropy Approximation, Variational Inference, Approximation Error Analysis

## 1 Introduction

Bayesian inference is a well-studied approach to equip a machine learning model with uncertainty estimation. Let $f(\,\cdot\,;w)$ be the base model that is a function parameterized by weights $w \in \mathbb{R}^m$, e.g., the neural network. Let $p(w)$ and $p(y|f(x;w))$ be the prior distribution of the weights and the likelihood of the model, respectively. For a supervised learning, let $D = \{(x_n, y_n)\}_{n=1}^N$ be a dataset where $x_n \in \mathbb{R}^{d_x}$ and $y_n \in \mathbb{R}^{d_y}$ are the input and output, respectively, and the input–output pair $(x_n, y_n)$ is independently identically distributed. The Bayesian posterior distribution $p(w|D)$ is formulated as

$$p(w|D) \propto p(w) \prod_{n=1}^N p(y_n|f(x_n;w)).$$

Unfortunately, the posterior distribution cannot be generally computed in a closed form in the case of nonlinear models such as neural networks, which are called Bayesian neural networks (BNNs) (MacKay, 1992; Neal, 2012). Therefore, the posterior distribution is often approximately obtained as a variational family using the variational inference (Hinton and Van Camp, 1993; Barber and Bishop, 1998; Bishop, 2006), which minimizes the Kullback–Leibler (KL) divergence between a posterior distribution and variational family. Although the unimodal Gaussian distribution is usually chosen as the variational family (Barber and Bishop, 1998; Opper and Archambeau, 2009), it hardly approximates multimodal distributions that are often assumed as posterior distributions of BNNs (see, e.g., Fort et al. (2019)).

In this paper, we employ the *Gaussian mixture* distribution (the superposition of Gaussian distributions) as our variational family because it is more flexible than the unimodal Gaussian distribution (see, e.g.,

Bacharoglou (2010)). A main difficulty of variational inference with the Gaussian mixture is how to approximate the entropy of the Gaussian mixture. Note that minimizing the KL divergence is equivalent to maximizing the *evidence lower bound* (ELBO), which consists of three terms: the expected log-likelihood, the cross-entropy between a variational family and prior distribution, and the entropy of a variational family. The expected log-likelihood term can be statistically calculated by some Monte Carlo sampling called a stochastic gradient variational Bayes method (SGVB) (Kingma and Welling, 2013; Rezende et al., 2014; Kingma et al., 2015; Ranganath et al., 2014; Titsias and Lázaro-Gredilla, 2014), and the cross-entropy term can be analytically calculated if the prior is chosen as a unimodal Gaussian. Although the above two terms are not problematic even when extending the variational family to the Gaussian mixture, the entropy term can no longer be analytically calculated. Hence, we approximate the entropy of the Gaussian mixture as the sum of the entropy of the unimodal Gaussian equation 3.5, which can be analytically calculated. This approximation has been originally proposed in Gal and Ghahramani (2016) based on the intuition that the true entropy tends to the approximated one when the dimension $m$ is large enough and means of the mixture are randomly distributed. Note that the situation where the dimension $m$ is large does not deviate from the BNN, which is one of typical applications of the variational inference, because the number of weights in neural networks is often large.

We theoretically analyze the approximation error between the true entropy and approximated one in order to reveal when our approximation works well. Specifically, we prove that the approximation error is controlled by the ratios of the distances between the means to the sum of the variances of the Gaussian mixture, and it converges to zero when the ratios go to infinity (Definition 4.1 and Theorem 4.2). This situation seems to be more likely to occur in higher dimensional parametric spaces because of the curse of dimensionality. Therefore, our result guarantees that our approximation works well in the BNN, for example. Furthermore, we also show the approximation error bound in the form of a probabilistic inequality (Corollary 4.3). This probabilistic inequality mathematically formulates and justifies the intuition in Gal and Ghahramani (2016).

The summary of our contributions is the following:

- We provide the approximation formula for the entropy of the Gaussian mixture in a closed form, which is suitable for the variational inference (Section 3).

- We analyze the approximation error between the entropy of the Gaussian mixture and the sum of the entropy of the unimodal Gaussian to theoretically justify the adopted approximation (Section 4).

## 2 Related work

A variety of approximations for the Bayesian posterior has been developed including the Laplace approximation (Mackay, 1992) and the Markov chain Monte Carlo method (Springenberg et al., 2016; Welling and Teh, 2011; Wenzel et al., 2020). In particular, there is a large body of literature on variational inference, where the case of the unimodal Gaussian or nonparametric factorized distribution has been studied well (see, e.g., Barber and Bishop (1998); Bishop (2006); Graves (2011); Louizos and Welling (2016); Opper and Archambeau (2009)). Variational inference with the Gaussian mixture was first studied in Jaakkola and Jordan (1998). On the entropy approximation, there are several studies (Huber et al., 2008; Bonilla et al., 2019; Zobay, 2014). The entropy approximation proposed in Huber et al. (2008) combines the Taylor approximation with the splitting method of Gaussian mixture components. As the order of Taylor series and the number of splitting go to infinity, the approximation error converges to zero (Huber et al., 2008, Theorem 1). However, the order and number are hyperparameters and it is not clear how to theoretically decide them. In Huber et al. (2008), the numerical experiment was performed on two-dimensional Gaussian mixtures, where it works well, whereas there is no experiment for the higher dimensional case. The entropy approximation adopted in Bonilla et al. (2019) is the lower bound of the true entropy based on Jensen's inequality. This approximation can be analytically calculated in the closed form, whereas there is no theoretical guarantee when it works well. The entropy approximation employed in Zobay (2014) includes the two dimensional integral, which cannot be backpropagated in the SGVB method. In our work, we use the tractable approximation formula in the SGVB method and provide a theoretical guarantee when it works well. In particular, it works better

in the high dimensional parametric space than other methods. As another problem, variational inference is sensitive to the choice of the prior as well as the variational family (see, e.g., MacKay (1992); Neal (1993)). To mitigate this problem, functional variational inference (Sun et al., 2018) and deterministic variational inference (Wu et al., 2019) have appeared in recent years. However, we do not focus on this problem since our aim is to obtain an executable and theoretically guaranteed variational inference.

Recently, deep ensembles (D'Angelo and Fortuin, 2021; Fort et al., 2019; Lakshminarayanan et al., 2017; Pearce et al., 2020; Rahaman and Thiery, 2021) have attracted attention as a simple method to estimate the uncertainty of neural networks. The idea of deep ensembles is to train individual deterministic neural networks[1] with random initialization and to predict by statistics of their outputs, which yields the sufficient predictive uncertainty in spite of lacking interpretation as probabilistic models. The deep ensemble is similar to our proposed method in the sense that it can capture the multimodality. Unfortunately, compared with deep ensembles, our method requires high computational cost because the Gaussian mixture includes many parameters, which may cause difficulty in application for real-time tasks. However, our method has a canonical statistical nature as Bayesian model, which is absent in the deep ensemble. As different approaches from ours, deep ensembles incorporating Bayesian interpretation have been developed, such as MultiSWAG (Maddox et al., 2019; Wilson and Izmailov, 2020) and deep ensembles via NTK (He et al., 2020).

## 3 Variational inference with Gaussian mixture

In this section, we briefly review the variational inference with the Gaussian mixture at first. Then, we provide the approximation formula for the entropy of the Gaussian mixture in a closed form, which is suitable for the variational inference. For the notation, see the beginning of Section 1.

The goal of variational inference is to minimize the Kullback-Leibler (KL) divergence between a variational family $q_\theta(w)$ and posterior distribution $p(w|D)$ given by

$$D_{\mathrm{KL}}(q_\theta(w) \,||\, p(w|D)) := - \int q_\theta(w) \log \left( \frac{p(w|D)}{q_\theta(w)} \right) dw,$$

which is equivalent to maximizing the evidence lower bound (ELBO) given by

$$\mathcal{L}(\theta) := L(\theta) + \int q_\theta(w) \log(p(w)) \, dw + H[q_\theta], \qquad (3.1)$$

(see, e.g., Barber and Bishop (1998); Bishop (2006); Hinton and Van Camp (1993)). The first term of equation 3.1 is the expected log-likelihood given by

$$L(\theta) := \sum_{n=1}^{N} E_{q_\theta(w)}[\log p(y_n | f(x_n; w))],$$

the second term is the cross-entropy between a variational family $q_\theta(w)$ and prior distribution $p(w)$, and the third term is the entropy of $q_\theta(w)$ given by

$$H[q_\theta] := - \int q_\theta(w) \log(q_\theta(w)) \, dw. \qquad (3.2)$$

In this study, we choose an unimodal Gaussian distribution as a prior, that is, $p(w) = \mathcal{N}(w|\mu_0, \Sigma_0)$, and we choose a Gaussian mixture distribution as a variational family, that is,

$$q_\theta(w) = \sum_{k=1}^{K} \pi_k \mathcal{N}(w|\mu_k, \Sigma_k), \quad \theta = (\pi_k, \mu_k, \Sigma_k)_{k=1}^{K},$$

where $K \in \mathbb{N}$ is the number of mixture components, and $\pi_k \in (0, 1]$ are mixing coefficients constrained by $\sum_{k=1}^{K} \pi_k = 1$. Here, $\mathcal{N}(w|\mu_k, \Sigma_k)$ is the Gaussian distribution with a mean $\mu_k \in \mathbb{R}^m$ and covariance matrix

---

[1]The terminology "deterministic" is used to distinguish the ordinal neural network from the BNN.

$\Sigma_k \in \mathbb{R}^{m \times m}$, that is,

$$\mathcal{N}(w|\mu_k, \Sigma_k) = \frac{1}{\sqrt{(2\pi)^m |\Sigma_k|}} \exp\left(-\frac{1}{2} \|w - \mu_k\|_{\Sigma_k}^2\right),$$

where $|\Sigma_k|$ is the determinant of matrix $\Sigma_k$, and $\|x\|_\Sigma^2 := x \cdot (\Sigma^{-1} x)$ for a vector $x \in \mathbb{R}^m$ and a positive definite matrix $\Sigma \in \mathbb{R}^{m \times m}$.

In the following, we investigate the ingredients in equation 3.1. The expected log-likelihood $L(\theta)$ is analytically intractable due to the nonlinearity of the function $f(x; w)$. To overcome this difficulty, we follow the stochastic gradient variational Bayes (SGVB) method (Kingma and Welling, 2013; Kingma et al., 2015; Rezende et al., 2014), which employs the reparametric trick and minibatch-based Monte Carlo sampling. Let $S \subset D$ be a minibatch set with minibatch size $M$. By reparameterizing weights as $w = \Sigma_k^{1/2} \varepsilon + \mu_k$, we can rewrite the expected log-likelihood $L(\theta)$ as

$$
\begin{aligned}
L(\theta) &= \sum_{n=1}^{N} \sum_{k=1}^{K} \pi_k \int \mathcal{N}(w|\mu_k, \Sigma_k) \log p(y_n | f(x_n; w)) \, dw \\
&= \sum_{n=1}^{N} \sum_{k=1}^{K} \pi_k \int \mathcal{N}(\varepsilon|0, I) \log p(y_n | f(x_n; \Sigma_k^{\frac{1}{2}} \varepsilon + \mu_k)) \, d\varepsilon.
\end{aligned}
$$

By minibatch-based Monte Carlo sampling, we obtain the following unbiased estimator $\widehat{L}^{\mathrm{SGVB}}(\theta)$ of the expected log-likelihood $L(\theta)$ as

$$
\begin{aligned}
L(\theta) &\approx \widehat{L}^{\mathrm{SGVB}}(\theta) \\
&:= \sum_{k=1}^{K} \pi_k \frac{N}{M} \sum_{i \in S} \log p(y_i | f(x_i; \Sigma_k^{\frac{1}{2}} \varepsilon + \mu_k)),
\end{aligned}
\tag{3.3}
$$

where we employ noise sampling $\varepsilon \sim \mathcal{N}(0, I)$ once per mini-batch sampling (Kingma et al. (2015); Titsias and Lázaro-Gredilla (2014)). On the other hand, the cross-entropy between a Gaussian mixture and unimodal Gaussian distribution can be analytically computed as

$$\int q_\theta(w) \log(p(w)) \, dw = -\sum_{k=1}^{K} \frac{\pi_k}{2} \left\{ m \log 2\pi + \log |\Sigma_0| + \mathrm{Tr}(\Sigma_0^{-1} \Sigma_k) + \|\mu_k - \mu_0\|_{\Sigma_0}^2 \right\}. \tag{3.4}$$

However, the entropy term $H[q_\theta]$ cannot be analytically computed when the variational family $q_\theta(w)$ is a Gaussian mixture. Therefore, we employ the approximation as

$$
\begin{aligned}
H[q_\theta] &\approx \widetilde{H}[q_\theta] \\
&:= -\sum_{k=1}^{K} \pi_k \int \mathcal{N}(w|\mu_k, \Sigma_k) \log \left(\pi_k \mathcal{N}(w|\mu_k, \Sigma_k)\right) dw \\
&= \frac{m}{2} + \frac{m}{2} \log 2\pi + \frac{1}{2} \sum_{k=1}^{K} \pi_k \log |\Sigma_k| - \sum_{k=1}^{K} \pi_k \log \pi_k.
\end{aligned}
\tag{3.5}
$$

The entropy approximation $\widetilde{H}[q_\theta]$ is the sum of the entropy of the unimodal Gaussian distribution, which can be analytically computed. This approximation has been originally proposed in Gal and Ghahramani (2016) based on the intuition that the true entropy $H[q_\theta]$ tends to the approximated one $\widetilde{H}[q_\theta]$ when the dimension $m$ is large enough and means $\mu_k$ are randomly distributed. In Section 4, we will analyze the approximation error $|H[q_\theta] - \widetilde{H}[q_\theta]|$ and theoretically justify this approximation.

In summary, we approximate the ELBO $\mathcal{L}(\theta)$ using equation 3.3, equation 3.4, and equation 3.5 as

$$
\begin{aligned}
\mathcal{L}(\theta) &\approx \widehat{\mathcal{L}}(\theta) \\
&:= \widehat{L}^{\mathrm{SGVB}}(\theta) + \int q_\theta(w) \log(p(w)) \, dw + \widetilde{H}[q_\theta] \\
&= \sum_{k=1}^{K} \pi_k \left( \widehat{\mathcal{L}}(\mu_k, \Sigma_k) - \log \pi_k \right),
\end{aligned}
\tag{3.6}
$$

where $\widehat{\mathcal{L}}(\mu_k, \Sigma_k)$ are ELBOs of the unimodal Gaussian distributions $\mathcal{N}(w|\mu_k, \Sigma_k)$ given by

$$
\begin{aligned}
\widehat{\mathcal{L}}(\mu_k, \Sigma_k) :=& \frac{N}{M} \sum_{i \in S} \log p(y_i | f(x_i; \Sigma_k^{\frac{1}{2}} \varepsilon_S + \mu_k)) \\
&- \frac{1}{2} \left\{ m \log 2\pi + \log |\Sigma_0| + \mathrm{Tr}(\Sigma_0^{-1} \Sigma_k) + \|\mu_k - \mu_0\|_{\Sigma_0}^2 \right\} \\
&+ \frac{m}{2}(1 + \log 2\pi) + \frac{1}{2} \log |\Sigma_k|.
\end{aligned}
$$

## 4 Error analysis for the entropy approximation

In this section, we analyze the approximation error $|H[q_\theta] - \widetilde{H}[q_\theta]|$ to theoretically justify the entropy approximation (3.5). We introduce the following notation to state results.

**Definition 4.1.** *For two Gaussian distributions* $\mathcal{N}(\mu_k, \Sigma_k)$ *and* $\mathcal{N}(\mu_{k'}, \Sigma_{k'})$, *we define* $\alpha_{\{k,k'\}}$ *by*

$$
\alpha_{\{k,k'\}} := \max_{\{x \in \mathbb{R}^m : \|x - \mu_k\|_{\Sigma_k} < \alpha\} \cap \{x \in \mathbb{R}^m : \|x - \mu_{k'}\|_{\Sigma_{k'}} < \alpha\} = \varnothing} \alpha,
$$

*and* $\alpha_{k,k'}$ *by*

$$
\alpha_{k,k'} := \frac{\|\mu_k - \mu_{k'}\|_{\Sigma_k}}{1 + \|\Sigma_k^{-\frac{1}{2}} \Sigma_{k'}^{\frac{1}{2}}\|_{\mathrm{op}}},
\tag{4.1}
$$

*where* $k, k' \in [K] := \{1, \dots, K\}$ *and* $\|\cdot\|_{\mathrm{op}}$ *is the operator norm (i.e., the largest singular value).*

**Interpretation of $\alpha$:** We can interpret that $\alpha_{\{k,k'\}}$ and $\alpha_{k,k'}$ measure distances of two Gaussian distributions in some sense respectively. Here notice that $\alpha_{\{k,k'\}}$ is always equal to $\alpha_{\{k',k\}}$, but $\alpha_{k,k'}$ is not always equal to $\alpha_{k',k}$.

Figure 1 shows the geometric interpretation of $\alpha_{k,k'}$ in the isotropic case, that is, $\Sigma_k = \sigma_k^2 I$ and $\Sigma_{k'} = \sigma_{k'}^2 I$. In this case, $\alpha_{k,k'}$ has a symmetric form with respect to $k, k'$ as

$$
\alpha_{\{k,k'\}} = \alpha_{k,k'} = \alpha_{k',k} = \frac{|\mu_k - \mu_{k'}|}{\sigma_k + \sigma_{k'}}.
$$

Here, the volume of $\mathcal{N}(\mu_k, \sigma_k^2 I)$ on $B(\mu_k, \alpha\sigma_k)$ is equal to the volume of $\mathcal{N}(\mu_{k'}, \sigma_{k'}^2 I)$ on $B(\mu_{k'}, \alpha\sigma_{k'})$, where $B(\mu, \sigma) := \{x \in \mathbb{R}^m : |x - \mu| < \sigma\}$ for $\mu \in \mathbb{R}^m$ and $\sigma > 0$. If all distances $|\mu_k - \mu_{k'}|$ between the means go to infinity, or all variances $\sigma_k$ go to zero, etc., then $\alpha_{k,k'}$ go to infinity for all pairs of $k, k'$. Furthermore, if all means $\mu_k$ are normally distributed (variances $\sigma_k$ are fixed), then an expected value of $\alpha_{k,k'}^2$ is in proportion to the dimension $m$. That intuitively means that the expected value of $\alpha_{k,k'}$ become large as the dimension $m$ of parameter increases.

These intuitions also hold in anisotropic case. From definition (4.1), we have

$$
\alpha_{k,k'} + \alpha_{k,k'} \sigma = |\Sigma_k^{-\frac{1}{2}} \mu_k - \Sigma_k^{-\frac{1}{2}} \mu_{k'}|,
$$

where $\sigma = \|\Sigma_k^{-\frac{1}{2}} \Sigma_{k'}^{\frac{1}{2}}\|_{\mathrm{op}}$ is the largest singular value of $\Sigma_k^{-\frac{1}{2}} \Sigma_{k'}^{\frac{1}{2}}$. This shows that the circumsphere of $\{x \in \mathbb{R}^m : \|x - \Sigma_k^{-\frac{1}{2}} \mu_{k'}\|_{\Sigma_k^{-1} \Sigma_k'} < \alpha_{k,k'}\}$ circumscribes $B(\Sigma_k^{-\frac{1}{2}} \mu_k, \alpha)$ (see Figure 2 right). Moreover

by coordinate transformation $x \to \Sigma_k^{\frac{1}{2}} x$, we can interpret that $\alpha_{k,k'}$ is a distance of $\mathcal{N}(\mu_k, \Sigma_k)$ and $\mathcal{N}(\mu_{k'}, \Sigma_{k'})$ from the perspective of $\Sigma_k$ (see Figure 2 left), and $\alpha_{k,k'}$ gives a concrete form of $\alpha$ satisfying $\{\|x - \mu_k\|_{\Sigma_k} < \alpha\} \cap \{\|x - \mu_{k'}\|_{\Sigma_{k'}} < \alpha\} = \varnothing$, that is, $\alpha_{k,k'} \leq \alpha_{\{k,k'\}}$ (Remark A.4).

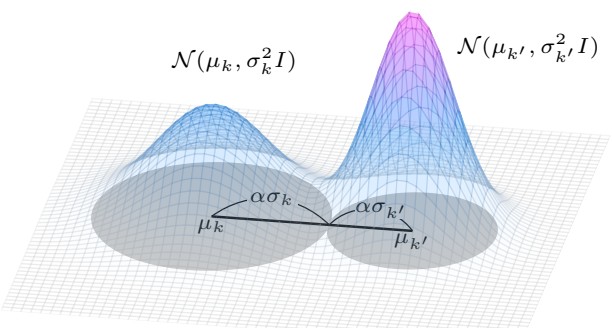

Figure 1: Illustration of $\alpha = \alpha_{\{k,k'\}}$ ($m = 2$, isotropic)

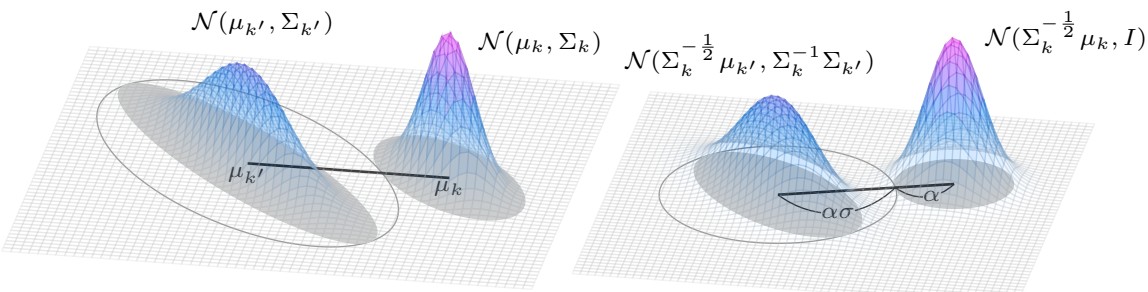

Figure 2: Illustration of $\alpha = \alpha_{k,k'}$ ($m = 2$, anisotropic)

### 4.1 General covariance case

We study the error $|H[q_\theta] - \widetilde{H}[q_\theta]|$ for general covariance matrices $\Sigma_k$. First, we give the following upper and lower bounds for the error.

**Theorem 4.2.** *For $s \in (0, 1)$,*

$$
\sum_{k=1}^{K} \sum_{k' \neq k} \frac{\pi_k \pi_{k'}}{1 - \pi_k} c_{k,k'} \log\left(1 + \frac{1 - \pi_k}{\pi_k} \frac{|\Sigma_k|^{\frac{1}{2}}}{\max_l |\Sigma_l|^{\frac{1}{2}}} \exp\left(-\frac{\left(1 + \|\Sigma_{k'}^{-\frac{1}{2}} \Sigma_k^{\frac{1}{2}}\|_{\mathrm{op}}\right)^2}{2} \alpha_{k',k}^2\right)\right)
$$

$$
\leq \left|H[q_\theta] - \widetilde{H}[q_\theta]\right| \leq \min\left\{\frac{K}{2}, \frac{2}{(1-s)^{\frac{m}{4}}} \sum_{k=1}^{K} \sum_{k' \neq k} \sqrt{\pi_k \pi_{k'}} \exp\left(-\frac{s\alpha_{k,k'}^2}{4}\right)\right\},
\tag{4.2}
$$

*where the coefficient $c_{k,k'}$ is defined by*

$$
c_{k,k'} := \frac{1}{\sqrt{(2\pi)^m}} \int_{\mathbb{R}^m_{k,k'}} \exp\left(-\frac{|y|}{2}\right) dy \geq 0,
$$

*and the set $\mathbb{R}^m_{k,k'}$ is defined by*

$$
\mathbb{R}^m_{k,k'} := \left\{y \in \mathbb{R}^m : \begin{array}{l} y \cdot y \geq (\Sigma_k^{\frac{1}{2}} \Sigma_{k'}^{-1} \Sigma_k^{\frac{1}{2}} y) \cdot y, \\ y \cdot (\Sigma_k^{\frac{1}{2}} \Sigma_{k'}^{-1} (\mu_{k'} - \mu_k)) \geq 0 \end{array}\right\}.
$$

*Moreover, the same upper bound holds for $\alpha_{\{k,k'\}}$ instead of $\alpha_{k,k'}$:*

$$\left| |H[q_\theta] - \widetilde{H}[q_\theta]| \right| \le \min\left\{ \frac{K}{2}, \frac{2}{(1-s)^{\frac{m}{4}}} \sum_{k=1}^{K} \sum_{k' \neq k} \sqrt{\pi_k \pi_{k'}} \exp\left( -\frac{s\alpha_{\{k,k'\}}^2}{4} \right) \right\}.$$

The proof is given in Appendix A.1. Theorem 4.2 implies following facts:

(i) The error is bounded from above by a constant $K/2$, which does not blow up with respect to the mean $\mu_k$, the covariance $\Sigma_k$, and the dimension $m$ if the number of mixture components $K$ is fixed.

(ii) According to another upper bound, the error exponentially decays to zero as $\alpha_{k,k'}$ go to infinity for all pairs $k, k' \in [K]$ with $k \neq k'$.

(iii) According to the lower bound, the condition that $\alpha_{k,k'}$ go to infinity for all $k, k' \in [K]$ with $c_{k,k'} > 0$ is necessary for the error $|H[q_\theta] - \widetilde{H}[q_\theta]|$ to converge to zero. Here, either $c_{k,k'}$ or $c_{k',k}$ is always positive (Remark A.6).

In addition, we provide the following probabilistic inequality for the error.

**Corollary 4.3.** *Assume $\{\mu_k\}_k$ and $\{\Sigma_k\}_k$ such that*

$$\frac{\Sigma_k^{-\frac{1}{2}}(\mu_k - \mu_{k'})}{1 + \|\Sigma_k^{-\frac{1}{2}}\Sigma_{k'}^{\frac{1}{2}}\|_{\mathrm{op}}} \sim \mathcal{N}(0, c^2 I), \tag{4.3}$$

*for all pairs $k, k' \in [K]$ ($k \neq k'$), that is, the left-hand side follows a Gaussian distribution with zero mean and an isotropic covariance matrix $c^2 I$. Then, for $\varepsilon > 0$ and $s \in (0,1)$,*

$$P\left( \left| H[q_\theta] - \widetilde{H}[q_\theta] \right| \ge \varepsilon \right) \le \frac{2(K-1)}{\varepsilon} \left( \sqrt{1-s}\left(1 + \frac{sc^2}{2}\right) \right)^{-\frac{m}{2}}. \tag{4.4}$$

The proof is given in Appendix A.2. The idea of the proof is that an expected value of $\alpha_{k,k'}^2$ is $c^2 m$ by the assumption (4.3), which implies that the upper bound in (4.2) converges to zero as the dimension $m$ goes to infinity. Corollary 4.3 justifies Gal and Ghahramani (2016, Proposition 1 in Appendix A), which formally mentions that $H[q_\theta]$ tends to $\widetilde{H}[q_\theta]$ when means $\mu_k$ are normally distributed, all elements of covariance matrices $\Sigma_k$ do not depend on $m$, and $m$ is large enough. In fact, the right hand side of equation 4.4 converges to zero as $m \to \infty$ for some $s \in (0,1)$ if $c > 1$.

We also study the derivative of the error $|H[q_\theta] - \widetilde{H}[q_\theta]|$ with respect to learning parameters $\theta = (\pi_k, \mu_k, \Sigma_k)_{k=1}^{K}$. For simplicity, we denote by

$$\Gamma_k := \Sigma_k^{\frac{1}{2}}.$$

We give the following upper bounds for the derivative of the error.

**Theorem 4.4.** *For $k \in [K]$, $p, q \in [m]$, and $s \in (0,1)$,*

(i) $\left| \dfrac{\partial}{\partial \mu_{k,p}} \left( H[q_\theta] - \widetilde{H}[q_\theta] \right) \right|$

$$\le \frac{2}{(1-s)^{\frac{m+2}{4}}} \sum_{k' \neq k} \sqrt{\pi_k \pi_{k'}} \left( \|\Gamma_{k'}^{-1}\|_1 + \|\Gamma_k^{-1}\|_1 \right) \exp\left( -\frac{s\max(\alpha_{k,k'}, \alpha_{k',k})^2}{4} \right), \tag{4.5}$$

(ii) $\left| \dfrac{\partial}{\partial \gamma_{k,pq}} \left( H[q_\theta] - \widetilde{H}[q_\theta] \right) \right|$

$$\le \frac{6}{(1-s)^{\frac{m+4}{4}}} \sum_{k' \neq k} \sqrt{\pi_k \pi_{k'}} \left( 2|\Gamma_k|^{-1}|\Gamma_{k,pq}| + \|\Gamma_k^{-1}\|_1 + \|\Gamma_{k'}^{-1}\|_1 \right) \exp\left( -\frac{s\max(\alpha_{k,k'}, \alpha_{k',k})^2}{4} \right), \tag{4.6}$$

(iii) $\left| \dfrac{\partial}{\partial \pi_k} \left( H[q_\theta] - \widetilde{H}[q_\theta] \right) \right| \le \dfrac{8}{(1-s)^{\frac{m}{4}}} \sum_{k' \neq k} \sqrt{\frac{\pi_{k'}}{\pi_k}} \exp\left( -\frac{s\max(\alpha_{k,k'}, \alpha_{k',k})^2}{4} \right), \tag{4.7}$

where $\mu_{k,p}$ and $\gamma_{k,pq}$ is the $p$-th and $(p,q)$-th components of vector $\mu_k$ and matrix $\Gamma_k$, respectively, and $\|\cdot\|_1$ is the entry-wise matrix 1-norm, and $|\Gamma_{k,pq}|$ is the determinant of the $(m-1) \times (m-1)$ matrix that results from deleting $p$-th row and $q$-th column of matrix $\Gamma_k$.

The proof is given in Appendix A.3. We observe that even in the derivative of the error, the upper bound exponentially decays to zero as $\alpha_{k,k'}$ go to infinity for all pairs $k, k' \in [K]$ with $k \neq k'$. We can also show that if means $\mu_k$ are normally distributed with certain large standard deviation $c$, then the probabilistic inequality like Corollary 4.3 that the bound converges to zero as $m$ goes to infinity is obtained.

### 4.2  Coincident covariance case

We study the error $|H[q_\theta] - \widetilde{H}[q_\theta]|$ for coincident covariance matrices, that is,

$$\Sigma_k = \Sigma \quad \text{for all } k \in [K],$$

where $\Sigma \in \mathbb{R}^{m \times m}$ is a positive definite matrix. In this case, $\alpha_{k,k'}$ have the form as

$$\alpha_{k,k'} = \frac{\|\mu_k - \mu_{k'}\|_\Sigma}{2}, \quad k, k' \in [K].$$

First, we show the following explicit form of the true entropy $H[q_\theta]$.

**Proposition 4.5.** *Let $m \geq K \geq 2$. Then,*

$$H[q_\theta] = \widetilde{H}[q_\theta] - \sum_{k=1}^{K} \frac{\pi_k}{(2\pi)^{\frac{K-1}{2}}} \int_{\mathbb{R}^{K-1}} \exp\left(-\frac{|v|^2}{2}\right) \log\left(1 + \sum_{k' \neq k} \frac{\pi_{k'}}{\pi_k} \exp\left(\frac{|v|^2 - |v - u_{k',k}|^2}{2}\right)\right) dv, \quad (4.8)$$

*where $u_{k',k} := [R_k \Sigma^{-1/2}(\mu_{k'} - \mu_k)]_{1:K-1} \in \mathbb{R}^{K-1}$ and $R_k \in \mathbb{R}^{m \times m}$ is some rotation matrix such that*

$$R_k \Sigma^{-\frac{1}{2}}(\mu_{k'} - \mu_k) \in \text{span}\{e_1, \cdots, e_{K-1}\}, \quad k' \in [K].$$

*Here, $\{e_i\}_{i=1}^{K-1}$ is the standard basis in $\mathbb{R}^{K-1}$, and $u_{1:K-1} := (u_1, \ldots, u_{K-1})^T \in \mathbb{R}^{K-1}$ for $u = (u_1, \ldots, u_m)^T \in \mathbb{R}^m$.*

The special case $K = 2$ of (4.8) can be found in Zobay (2014, Appendix A). The proof is given in Appendix B.1. Using Proposition 4.5, we have the following upper and lower bounds for the error.

**Theorem 4.6.** *Let $m \geq K \geq 2$. Then, for $s \in (0,1)$,*

$$\frac{1}{2} \sum_{k=1}^{K} \sum_{k' \neq k} \frac{\pi_k \pi_{k'}}{1 - \pi_k} \log\left(1 + \frac{1 - \pi_k}{\pi_k} \exp(-2\alpha_{k',k}^2)\right)$$

$$\leq \left|H[q_\theta] - \widetilde{H}[q_\theta]\right| \leq \frac{2}{(1-s)^{\frac{K-1}{4}}} \sum_{k=1}^{K} \sum_{k' \neq k} \sqrt{\pi_k \pi_{k'}} \exp\left(-\frac{s\alpha_{k,k'}^2}{4}\right).$$

*Moreover, the same upper bound holds for $\alpha_{\{k,k'\}}$ instead of $\alpha_{k,k'}$.*

The proof is given in Appendix B.2. Theorem 4.6 implies that, in the coincident covariance matrices case, the condition that $\alpha_{k,k'}$ go to infinity for all $k, k' \in [K]$ with $k \neq k'$ is necessary and sufficient for the error $|H[q_\theta] - \widetilde{H}[q_\theta]|$ to converge to zero. The upper bound of Theorem 4.6 is sharper than that of Theorem 4.2 because there are no terms that blow up as $m \to \infty$ such as $(1-s)^{-m/4}$ in (4.2). This sharper bound leads to the following probabilistic inequality for the error.

**Corollary 4.7.** *Let $m \geq K \geq 2$. Assume $\{\mu_k\}_k$ and $\Sigma$ such that*

$$\frac{\Sigma^{-\frac{1}{2}}(\mu_k - \mu_{k'})}{2} \sim \mathcal{N}(0, c^2 I),$$

*for all pairs $k, k' \in [K]$ $(k \neq k')$. Then, for $\varepsilon > 0$ and $s \in (0,1)$,*

$$P\left(\left|H[q_\theta] - \widetilde{H}[q_\theta]\right| \geq \varepsilon\right) \leq \frac{2(K-1)}{\varepsilon(1-s)^{\frac{K-1}{4}}} \left(1 + \frac{sc^2}{2}\right)^{-\frac{m}{2}}.$$

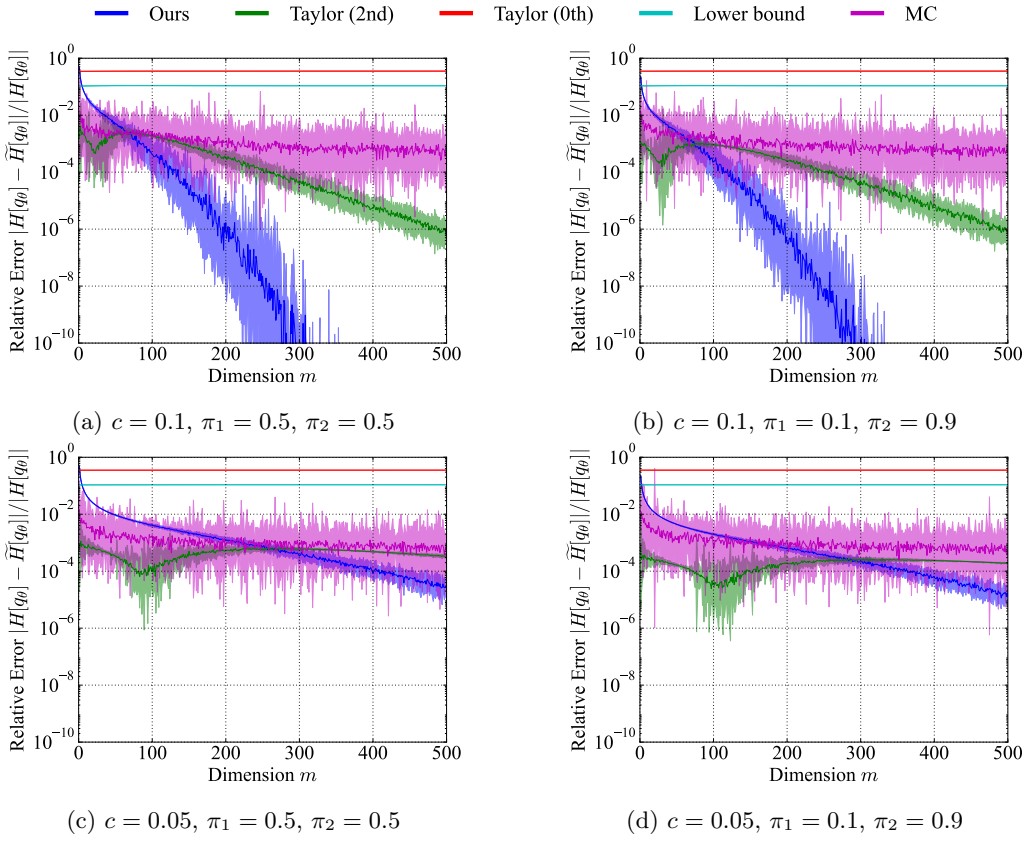

Figure 3: Relative error $|H[q_\theta] - \widetilde{H}_*[q_\theta]|/|H[q_\theta]|$ for the true entropy $H[q_\theta]$ and the approximation ones $\widetilde{H}_*[q_\theta]$. Each line indicates the mean value for 500 samples and filled region indicates the min–max interval. $c$ denotes the same symbol in Corollary 4.7. Methods of Ours, Taylor (2nd), Taylor (0th), and Lower bound denote $\widetilde{H}_{\text{ours}}[q_\theta]$, $\widetilde{H}_{\text{Huber(2)}}[q_\theta]$, $\widetilde{H}_{\text{Huber(0)}}[q_\theta]$, and $\widetilde{H}_{\text{Bonilla}}[q_\theta]$ in Appendix C.1, respectively. A method of MC denotes the Monte Carlo integration with 1000 sampling points.

The proof is given in Appendix B.3. Note that, in Corollary 4.7, the assumption $c > 1$ is not necessary anymore for the zero convergence, which is necessary in Corollary 4.3.

## 5 Experiments

We conducted some experiments to support the effectiveness of our approximation formula equation 3.5. We claim that our approximation has an advantage in higher dimension and the variational inference with our approximation works well.

### 5.1 Numerical analysis for the approximation error

We numerically examined approximation capabilities of our formula equation 3.5 compared with Huber et al. (2008), Bonilla et al. (2019), and the Monte Carlo integration. Generally, we cannot compute the entropy equation 3.2 in a closed form. Therefore, we restricted the setting of the experiment to the case for the coincident covariance matrices (Section 4.2), in particular $\Sigma = I$, and minimal number of the mixture components $K = 2$, where we obtained more tractable formula for the entropy equation C.4. In this setting, we investigated the relative error between the entropy and each approximation method, see Figure 3. The details for the experimental setting and exact formulas for each method are shown in Appendix C.1.

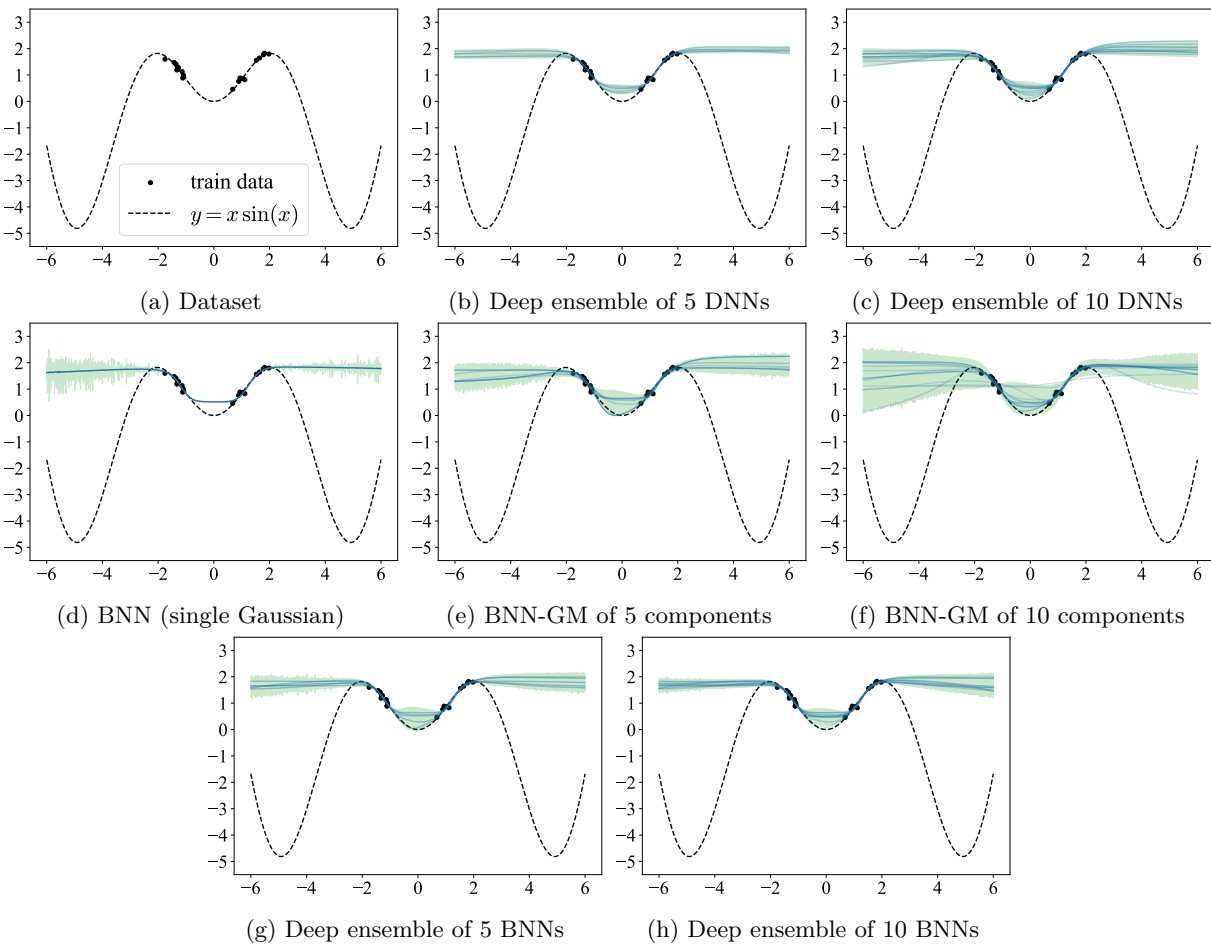

Figure 4: Uncertainty estimation on the toy 1D regression task. Each line indicates one component of the ensemble or mixture, and the filled region indicates the mean value of the prediction with the standard deviation$\times(\pm2)$. As for BNN-GMs, the stronger color intensity of the line, the larger mixture coefficients of the model.

From the result in Figure 3, we can observe the following. First, the relative error of ours shows faster decay than others in higher dimension $m$. Therefore, our approximation formula equation 3.5 has an advantage in higher dimension. Second, the graph for ours scales in the $x$-axis as $c$ scales, which is consistent with the expression of the upper bound in Corollary 4.7. Finally, ours is robust against varying mixing coefficients, which cannot be explained by Corollary 4.7. Note that we can hardly conduct a similar experiment for $K > 2$ because we cannot prepare the tolerant ground truth of the entropy. For example, even the Monte Carlo integration is not suitable for the ground truth already in the case for $K = 2$ due to its large relative error around $10^{-3}$.

## 5.2 BNN with Gaussian Mixture

We applied the proposed variational inference to a BNN whose posterior was modeled by the Gaussian mixture, which we call *BNN-GM* in the following. We conducted the toy 1 D regression task (He et al., 2020) to observe the uncertainty estimation capability of the BNN-GM. In particular, we observed that the BNN-GM could capture larger uncertainty than the deep ensemble (Lakshminarayanan et al., 2017). The task was to learn a curve $y = x \sin(x)$ from a train dataset that consisted of 20 points sampled from a noised curve $y = x \sin(x) + \varepsilon$, $\varepsilon \sim \mathcal{N}(0, 0.1^2)$. Refer to Appendix C.2 for the detail of implementations. We

compared the BNN-GM with the deep ensemble of DNNs, the BNN with the single unimodal Gaussian, and the deep ensemble of BNNs with the single unimodal Gaussian, see Figure 4.

From the result in Figure 4, we can observe the following. First, every method can represent uncertainty on the area where train data do not exist. However, the BNN with single unimodal Gaussian can represent smaller uncertainty than other methods (see around $x = 0$). Second, as increasing the number of components, the BNN-GM can represent larger uncertainty than the deep ensemble of DNNs or BNNs. Therefore, there is a qualitative difference in uncertainty estimation between BNN-GMs and deep ensembles. Finally, the BNN-GM of 10 components has weak learners with small mixture coefficients. We suppose that this phenomenon is caused by the entropy regularization for the Gaussian mixture. Note that we do not claim the superiority of BNN-GMs to deep ensembles.

## 6 Limitations and future works

The limitations and future work are as follows:

- There is an unsolved problem on the standard deviation $c$ in (4.3) of Corollary 4.3. According to this corollary, the approximation error almost surely converges to zero as $m \to \infty$ if we take $c > 1$ (the discussion after Corollary 4.3). However, it is unsolved whether the condition $c > 1$ is optimal or not for the convergence. According to Corollary 4.7, the condition $c > 1$ can be removed in the particular case $\Sigma_k = \Sigma$ for all $k \in [K]$.

- Our entropy approximation (3.5) is valid only when $\alpha_{k,k'}$ are large enough. However, since there are situations where $\alpha_{k,k'}$ are small, such as the low-dimensional latent space of a variational autoencoder (Kingma and Welling, 2013), it is worthwhile to propose an appropriate entropy approximation for small $\alpha_{k,k'}$.

- Compared with deep ensembles, our method requires high computational cost, which causes difficulty in application for real-time tasks. However, the Bayesian modeling, which is lacking in deep ensembles, is important in the sense that it gives a classical model the chance to sophisticate the learning theory. For example, it would be interesting to propose deep ensembles incorporating the concept of mixing coefficients $\pi_k$, which can be naturally trained in our Bayesian setting.

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

# Appendix

We define the Gaussian mixture distribution $q(x)$, its entropy $H[q]$, and approximated one $\widetilde{H}[q]$ as follows:

$$q(x) := \sum_{k=1}^{K} \pi_k \mathcal{N}(x|\mu_k, \Sigma_k),$$

$$H[q] := -\sum_{k=1}^{K} \pi_k \int_{\mathbb{R}^m} \mathcal{N}(x|\mu_k, \Sigma_k) \log \left( \sum_{k'=1}^{K} \pi_{k'} \mathcal{N}(x|\mu_{k'}, \Sigma_{k'}) \right) dx,$$

$$\widetilde{H}[q] := -\sum_{k=1}^{K} \pi_k \int_{\mathbb{R}^m} \mathcal{N}(x|\mu_k, \Sigma_k) \log \left( \pi_k \mathcal{N}(x|\mu_k, \Sigma_k) \right) dx,$$

where $K \in \mathbb{N}$ is the number of mixture components and $\pi_k \in (0, 1]$ are the mixing coefficients with $\sum_{k=1}^{K} \pi_k = 1$. Here, $\mathcal{N}(x|\mu_k, \Sigma_k)$ is the Gaussian distribution with a mean $\mu_k \in \mathbb{R}^m$ and covariance matrix $\Sigma_k \in \mathbb{R}^{m \times m}$, that is,

$$\mathcal{N}(x|\mu_k, \Sigma_k) = \frac{1}{\sqrt{(2\pi)^m |\Sigma_k|}} \exp\left( -\frac{1}{2} \|x - \mu_k\|_{\Sigma_k}^2 \right),$$

where $\|x\|_{\Sigma}^2 := x \cdot (\Sigma^{-1} x)$ for a vector $x \in \mathbb{R}^m$ and a positive definite matrix $\Sigma \in \mathbb{R}^{m \times m}$. We also define $\alpha_{\{k,k'\}}$ and $\alpha_{k,k'}$ by

$$\alpha_{\{k,k'\}} := \max_{\{x \in \mathbb{R}^m : \|x - \mu_k\|_{\Sigma_k} < \alpha\} \cap \{x \in \mathbb{R}^m : \|x - \mu_{k'}\|_{\Sigma_{k'}} < \alpha\} = \varnothing} \alpha,$$

$$\alpha_{k,k'} := \frac{\|\mu_k - \mu_{k'}\|_{\Sigma_k}}{1 + \|\Sigma_k^{-\frac{1}{2}} \Sigma_{k'}^{\frac{1}{2}}\|_{\mathrm{op}}},$$

where $k, k' \in [K] := \{1, \dots, K\}$ and $\|\cdot\|_{\mathrm{op}}$ is the operator norm (which is the largest singular value).

## A    General covariance case

### A.1    Proof of Theorem 4.2

**Theorem A.1** (Theorem 4.2 (I))**.**

$$\left| H[q] - \widetilde{H}[q] \right| \leq \frac{K}{2}.$$

*Proof.* Making the change of variables as $y = \Sigma_k^{-1/2}(x - \mu_k)$, we write

$$\left| H[q] - \widetilde{H}[q] \right|$$

$$= \sum_{k=1}^{K} \pi_k \int_{\mathbb{R}^m} \mathcal{N}(x|\mu_k, \Sigma_k) \left\{ \log \left( \sum_{k'=1}^{K} \pi_{k'} \mathcal{N}(x|\mu_{k'}, \Sigma_k) \right) - \log \left( \pi_k \mathcal{N}(x|\mu_k, \Sigma_k) \right) \right\} dx$$

$$= \sum_{k=1}^{K} \pi_k \int_{\mathbb{R}^m} \frac{1}{\sqrt{(2\pi)^m |\Sigma_k|}} \exp\left(-\frac{\|x-\mu_k\|_{\Sigma_k}^2}{2}\right)$$

$$\times \log\left(1 + \sum_{k' \neq k} \frac{\pi_{k'} |\Sigma_k|^{\frac{1}{2}}}{\pi_k |\Sigma_{k'}|^{\frac{1}{2}}} \exp\left(\frac{\|x-\mu_k\|_{\Sigma_k}^2 - \|x-\mu_{k'}\|_{\Sigma_{k'}}^2}{2}\right)\right) dx$$

$$= \sum_{k=1}^{K} \pi_k \int_{\mathbb{R}^m} \frac{1}{\sqrt{(2\pi)^m}} \exp\left(-\frac{|y|^2}{2}\right)$$

$$\times \log\left(1 + \sum_{k' \neq k} \frac{\pi_{k'} |\Sigma_k|^{\frac{1}{2}}}{\pi_k |\Sigma_{k'}|^{\frac{1}{2}}} \exp\left(\frac{|y|^2 - \left\|\Sigma_k^{\frac{1}{2}}\left(y - \Sigma_k^{-\frac{1}{2}}(\mu_{k'}-\mu_k)\right)\right\|_{\Sigma_{k'}}^2}{2}\right)\right) dy.$$

(A.1)

Using the inequality $\log(1+x) \leq \sqrt{x}$ $(x \geq 0)$ and the Cauchy-Schwarz inequality, we have

$$\left|H[q] - \widetilde{H}[q]\right|$$

$$\leq \sum_{k=1}^{K} \pi_k \int_{\mathbb{R}^m} \frac{1}{\sqrt{(2\pi)^m}} \exp\left(-\frac{|y|^2}{2}\right)$$

$$\times \sqrt{\sum_{k' \neq k} \frac{\pi_{k'} |\Sigma_k|^{\frac{1}{2}}}{\pi_k |\Sigma_{k'}|^{\frac{1}{2}}} \exp\left(\frac{|y|^2 - \left\|\Sigma_k^{\frac{1}{2}}\left(y - \Sigma_k^{-\frac{1}{2}}(\mu_{k'}-\mu_k)\right)\right\|_{\Sigma_{k'}}^2}{2}\right)} \, dy$$

$$= \sum_{k=1}^{K} \int_{\mathbb{R}^m} \frac{1}{(2\pi)^{\frac{m}{4}}} \exp\left(-\frac{|y|^2}{4}\right)$$

$$\times \sqrt{\frac{1}{(2\pi)^{\frac{m}{2}}} \sum_{k' \neq k} \pi_k \pi_{k'} \frac{|\Sigma_k|^{\frac{1}{2}}}{|\Sigma_{k'}|^{\frac{1}{2}}} \exp\left(\frac{-\left\|\Sigma_k^{\frac{1}{2}}\left(y - \Sigma_k^{-\frac{1}{2}}(\mu_{k'}-\mu_k)\right)\right\|_{\Sigma_{k'}}^2}{2}\right)} \, dy$$

$$\leq \sum_{k=1}^{K} \left(\int_{\mathbb{R}^m} \frac{1}{(2\pi)^{\frac{m}{2}}} \exp\left(-\frac{|y|^2}{2}\right) dy\right)^{\frac{1}{2}}$$

$$\times \underbrace{\left(\int_{\mathbb{R}^m} \frac{1}{(2\pi)^{\frac{m}{2}}} \sum_{k' \neq k} \pi_k \pi_{k'} \frac{|\Sigma_k|^{\frac{1}{2}}}{|\Sigma_{k'}|^{\frac{1}{2}}} \exp\left(\frac{-\left\|\Sigma_k^{\frac{1}{2}}\left(y - \Sigma_k^{-\frac{1}{2}}(\mu_{k'}-\mu_k)\right)\right\|_{\Sigma_{k'}}^2}{2}\right) dy\right)^{\frac{1}{2}}}_{=\left(\left(\sum_{k' \neq k}^{K} \pi_k \pi_{k'}\right) \int_{\mathbb{R}^m} \frac{1}{(2\pi)^{\frac{m}{2}}} \exp\left(-\frac{|z|^2}{2}\right) dz\right)^{\frac{1}{2}}}$$

$$= \sum_{k=1}^{K} \underbrace{\left(\sum_{k' \neq k} \pi_k \pi_{k'}\right)^{\frac{1}{2}}}_{=\sqrt{\pi_k(1-\pi_k)} \leq \frac{\pi_k + (1-\pi_k)}{2}} \leq \frac{K}{2}.$$

(A.2)

Therefore, we obtain Theorem A.1. $\qquad\square$

**Theorem A.2** (Theorem 4.2 (II)). *For $s \in (0,1)$,*

$$\left| H[q] - \widetilde{H}[q] \right| \leq \frac{2}{(1-s)^{\frac{m}{4}}} \sum_{k=1}^{K} \sum_{k' \neq k} \sqrt{\pi_k \pi_{k'}} \exp\left( -\frac{s \alpha_{k,k'}^2}{4} \right). \tag{A.3}$$

*Proof.* Using the first inequality in (A.2) and the inequality $\sqrt{\sum_i a_i} \leq \sum_i \sqrt{a_i}$, we decompose

$$\left| H[q] - \widetilde{H}[q] \right|$$

$$\leq \sum_{k=1}^{K} \sum_{k' \neq k} \sqrt{\pi_k \pi_{k'} \frac{|\Sigma_k|^{\frac{1}{2}}}{|\Sigma_{k'}|^{\frac{1}{2}}}} \int_{\mathbb{R}^m} \frac{1}{\sqrt{(2\pi)^m}} \exp\left( -\frac{|y|^2}{4} - \frac{\left\| \Sigma_k^{\frac{1}{2}}\left(y - \Sigma_k^{-\frac{1}{2}}(\mu_{k'} - \mu_k)\right) \right\|_{\Sigma_{k'}}^2}{4} \right) dy$$

$$= \sum_{k=1}^{K} \sum_{k' \neq k} \sqrt{\pi_k \pi_{k'} \frac{|\Sigma_k|^{\frac{1}{2}}}{|\Sigma_{k'}|^{\frac{1}{2}}}} \int_{|y|<\alpha_{k,k'}} \frac{1}{\sqrt{(2\pi)^m}} \exp\left( -\frac{|y|^2}{4} - \frac{\left\| \Sigma_k^{\frac{1}{2}}\left(y - \Sigma_k^{-\frac{1}{2}}(\mu_{k'} - \mu_k)\right) \right\|_{\Sigma_{k'}}^2}{4} \right) dy$$

$$+ \sum_{k=1}^{K} \sum_{k' \neq k} \sqrt{\pi_k \pi_{k'} \frac{|\Sigma_k|^{\frac{1}{2}}}{|\Sigma_{k'}|^{\frac{1}{2}}}} \int_{|y|>\alpha_{k,k'}} \frac{1}{\sqrt{(2\pi)^m}} \exp\left( -\frac{|y|^2}{4} - \frac{\left\| \Sigma_k^{\frac{1}{2}}\left(y - \Sigma_k^{-\frac{1}{2}}(\mu_{k'} - \mu_k)\right) \right\|_{\Sigma_{k'}}^2}{4} \right) dy$$

$$=: D^i + D^o.$$

Firstly, we evaluate the term $D^i$ of the integral over $|y| < \alpha_{k,k'}$. By the definition of $\alpha_{k,k'}$, we have

$$\left| \Sigma_k^{-\frac{1}{2}}(\mu_{k'} - \mu_k) \right| = \alpha_{k,k'}\left( 1 + \left\| \Sigma_k^{-\frac{1}{2}}\Sigma_{k'}^{\frac{1}{2}} \right\|_{\mathrm{op}} \right) > |y| + \alpha_{k,k'}\left\| \Sigma_k^{-\frac{1}{2}}\Sigma_{k'}^{\frac{1}{2}} \right\|_{\mathrm{op}}.$$

Then it follows from properties of $\|\cdot\|_{\Sigma_{k'}}, \|\cdot\|_{\mathrm{op}}$, and triangle inequality that

$$\left\| \Sigma_k^{\frac{1}{2}}\left(y - \Sigma_k^{-\frac{1}{2}}(\mu_{k'} - \mu_k)\right) \right\|_{\Sigma_{k'}} = \left| \Sigma_{k'}^{-\frac{1}{2}}\Sigma_k^{\frac{1}{2}}\left(y - \Sigma_k^{-\frac{1}{2}}(\mu_{k'} - \mu_k)\right) \right|$$

$$\geq \frac{\left| y - \Sigma_k^{-\frac{1}{2}}(\mu_{k'} - \mu_k) \right|}{\left\| \left(\Sigma_{k'}^{-\frac{1}{2}}\Sigma_k^{\frac{1}{2}}\right)^{-1} \right\|_{\mathrm{op}}} \geq \frac{\left| \Sigma_k^{-\frac{1}{2}}(\mu_{k'} - \mu_k) \right| - |y|}{\left\| \Sigma_k^{-\frac{1}{2}}\Sigma_{k'}^{\frac{1}{2}} \right\|_{\mathrm{op}}} > \alpha_{k,k'}. \tag{A.4}$$

From this inequality and the Cauchy-Schwarz inequality, it follows that for $s \in (0,1)$,

$$D^i = \sum_{k=1}^{K} \sum_{k' \neq k} \sqrt{\pi_k \pi_{k'} \frac{|\Sigma_k|^{\frac{1}{2}}}{|\Sigma_{k'}|^{\frac{1}{2}}}} \int_{|y|<\alpha_{k,k'}} \frac{1}{\sqrt{(2\pi)^m}}$$

$$\times \exp\left( -\frac{|y|^2}{4} - \frac{s\left\| \Sigma_k^{\frac{1}{2}}\left(y - \Sigma_k^{-\frac{1}{2}}(\mu_{k'} - \mu_k)\right) \right\|_{\Sigma_{k'}}^2}{4} - \frac{(1-s)\left\| \Sigma_k^{\frac{1}{2}}\left(y - \Sigma_k^{-\frac{1}{2}}(\mu_{k'} - \mu_k)\right) \right\|_{\Sigma_{k'}}^2}{4} \right) dy$$

$$\underset{\text{by (A.4)}}{\leq} \sum_{k=1}^{K} \sum_{k' \neq k} \sqrt{\pi_k \pi_{k'}} \exp\left( -\frac{s\alpha_{k,k'}^2}{4} \right) \int_{|y|<\alpha_{k,k'}} \frac{1}{(2\pi)^{\frac{m}{4}}} \exp\left( -\frac{|y|^2}{4} \right)$$

$$\times \sqrt{\frac{|\Sigma_k|^{\frac{1}{2}}}{|\Sigma_{k'}|^{\frac{1}{2}}}} \frac{1}{(2\pi)^{\frac{m}{4}}} \exp\left( -\frac{(1-s)\left\| \Sigma_k^{\frac{1}{2}}\left(y - \Sigma_k^{-\frac{1}{2}}(\mu_{k'} - \mu_k)\right) \right\|_{\Sigma_{k'}}^2}{4} \right) dy$$

$$\leq \sum_{k=1}^{K} \sum_{k' \neq k} \sqrt{\pi_k \pi_{k'}} \exp\left(-\frac{s\alpha_{k,k'}^2}{4}\right) \left(\int_{\mathbb{R}^m} \frac{1}{(2\pi)^{\frac{m}{2}}} \exp\left(-\frac{|y|^2}{2}\right) dy\right)^{\frac{1}{2}}$$

$$\times \left(\int_{\mathbb{R}^m} \frac{|\Sigma_k|^{\frac{1}{2}}}{|\Sigma_{k'}|^{\frac{1}{2}}} \frac{1}{(2\pi)^{\frac{m}{2}}} \exp\left(-\frac{(1-s)\left\|\Sigma_k^{\frac{1}{2}}\left(y - \Sigma_k^{-\frac{1}{2}}(\mu_{k'} - \mu_k)\right)\right\|_{\Sigma_{k'}}^2}{2}\right) dy\right)^{\frac{1}{2}}$$

$$= \sum_{k=1}^{K} \sum_{k' \neq k} \sqrt{\pi_k \pi_{k'}} \exp\left(-\frac{s\alpha_{k,k'}^2}{4}\right) \underbrace{\left(\int_{\mathbb{R}^m} \frac{1}{(2\pi)^{\frac{m}{2}}} \exp\left(-\frac{(1-s)|z|^2}{2}\right) dz\right)^{\frac{1}{2}}}_{=(1-s)^{-\frac{m}{4}}}$$

$$= \frac{1}{(1-s)^{\frac{m}{4}}} \sum_{k=1}^{K} \sum_{k' \neq k} \sqrt{\pi_k \pi_{k'}} \exp\left(-\frac{s\alpha_{k,k'}^2}{4}\right).$$

where we have used the change of variable as $z = \Sigma_{k'}^{-\frac{1}{2}} \Sigma_k^{\frac{1}{2}} \left(y - \Sigma_k^{-\frac{1}{2}}(\mu_{k'} - \mu_k)\right)$.

Secondly, we evaluate the term $D^o$ of the integral over $|y| > \alpha_{k,k'}$. In the same way as above, we have

$$D^o = \sum_{k=1}^{K} \sum_{k' \neq k} \sqrt{\pi_k \pi_{k'} \frac{|\Sigma_k|^{\frac{1}{2}}}{|\Sigma_{k'}|^{\frac{1}{2}}}} \int_{|y| > \alpha_{k,k'}} \frac{1}{\sqrt{(2\pi)^m}}$$

$$\times \exp\left(-\frac{s|y|^2}{4} - \frac{(1-s)|y|^2}{4} - \frac{\left\|\Sigma_k^{\frac{1}{2}}\left(y - \Sigma_k^{-\frac{1}{2}}(\mu_{k'} - \mu_k)\right)\right\|_{\Sigma_{k'}}^2}{4}\right) dy$$

$$\leq \sum_{k=1}^{K} \sum_{k' \neq k} \sqrt{\pi_k \pi_{k'}} \exp\left(-\frac{s\alpha_{k,k'}^2}{4}\right) \int_{|y| > \alpha_{k,k'}} \frac{1}{(2\pi)^{\frac{m}{4}}} \exp\left(-\frac{(1-s)|y|^2}{4}\right)$$

$$\times \sqrt{\frac{|\Sigma_k|^{\frac{1}{2}}}{|\Sigma_{k'}|^{\frac{1}{2}}}} \frac{1}{(2\pi)^{\frac{m}{4}}} \exp\left(-\frac{\left\|\Sigma_k^{\frac{1}{2}}\left(y - \Sigma_k^{-\frac{1}{2}}(\mu_{k'} - \mu_k)\right)\right\|_{\Sigma_{k'}}^2}{4}\right) dy$$

$$\leq \sum_{k=1}^{K} \sum_{k' \neq k} \sqrt{\pi_k \pi_{k'}} \exp\left(-\frac{s\alpha_{k,k'}^2}{4}\right) \left(\int_{\mathbb{R}^m} \frac{1}{(2\pi)^{\frac{m}{2}}} \exp\left(-\frac{(1-s)|y|^2}{2}\right) dy\right)^{\frac{1}{2}}$$

$$\times \left(\int_{\mathbb{R}^m} \frac{|\Sigma_k|^{\frac{1}{2}}}{|\Sigma_{k'}|^{\frac{1}{2}}} \frac{1}{(2\pi)^{\frac{m}{2}}} \exp\left(-\frac{\left\|\Sigma_k^{\frac{1}{2}}\left(y - \Sigma_k^{-\frac{1}{2}}(\mu_{k'} - \mu_k)\right)\right\|_{\Sigma_{k'}}^2}{2}\right) dy\right)^{\frac{1}{2}}$$

$$= \sum_{k=1}^{K} \sum_{k' \neq k} \sqrt{\pi_k \pi_{k'}} \exp\left(-\frac{s\alpha_{k,k'}^2}{4}\right) \underbrace{\left(\int_{\mathbb{R}^m} \frac{1}{(2\pi)^{\frac{m}{2}}} \exp\left(-\frac{(1-s)|y|^2}{2}\right) dy\right)^{\frac{1}{2}}}_{=(1-s)^{-\frac{m}{4}}}$$

$$= \frac{1}{(1-s)^{\frac{m}{4}}} \sum_{k=1}^{K} \sum_{k' \neq k} \sqrt{\pi_k \pi_{k'}} \exp\left(-\frac{s\alpha_{k,k'}^2}{4}\right).$$

Combining the estimates obtained now, we conclude (A.3). $\qquad\square$

**Theorem A.3** (Theorem 4.2 (III)). *For $s \in (0,1)$,*

$$\left| H[q] - \widetilde{H}[q] \right| \leq \frac{2}{(1-s)^{\frac{m}{4}}} \sum_{k=1}^{K} \sum_{k' \neq k} \sqrt{\pi_k \pi_{k'}} \exp\left( -\frac{s\alpha_{\{k,k'\}}^2}{4} \right).$$

*Proof.* The proof is almost same as Theorem A.2 except the evaluation (A.4). By the change of variables $y = \Sigma_k^{-\frac{1}{2}}(x - \mu_k)$, we have

$$\|x - \mu_k\|_{\Sigma_k} = \left| \Sigma_k^{-\frac{1}{2}}(x - \mu_k) \right| = |y|,$$

$$\|x - \mu_{k'}\|_{\Sigma_{k'}} = \left| \Sigma_{k'}^{-\frac{1}{2}}(x - \mu_{k'}) \right| = \left| \Sigma_{k'}^{-\frac{1}{2}} \Sigma_k^{\frac{1}{2}} \left( y - \Sigma_k^{-\frac{1}{2}}(\mu_{k'} - \mu_k) \right) \right| = \left\| \Sigma_k^{\frac{1}{2}} \left( y - \Sigma_k^{-\frac{1}{2}}(\mu_{k'} - \mu_k) \right) \right\|_{\Sigma_{k'}}.$$

From the definition of $\alpha_{\{k,k'\}}$,

$$\{y \in \mathbb{R}^m : |y| < \alpha\} \cap \left\{ y \in \mathbb{R}^m : \left\| \Sigma_k^{\frac{1}{2}} \left( y - \Sigma_k^{-\frac{1}{2}}(\mu_{k'} - \mu_k) \right) \right\|_{\Sigma_{k'}} < \alpha \right\} = \varnothing,$$

then if $|y| < \alpha_{\{k,k'\}}$, we obtain $\left\| \Sigma_k^{\frac{1}{2}} \left( y - \Sigma_k^{-\frac{1}{2}}(\mu_{k'} - \mu_k) \right) \right\|_{\Sigma_{k'}} \geq \alpha_{\{k,k'\}}$. $\qquad\square$

**Remark A.4.** *For any $k, k' \in [K]$,*

$$\alpha_{\{k,k'\}} \geq \alpha_{k,k'}.$$

*Proof.* When $x$ satisfies

$$\|x - \Sigma_k^{-\frac{1}{2}}\mu_{k'}\|_{\Sigma_k^{-1}\Sigma_{k'}} < \alpha_{k,k'},$$

since

$$\|x - \Sigma_k^{-\frac{1}{2}}\mu_{k'}\|_{\Sigma_k^{-1}\Sigma_{k'}} = \left| \Sigma_{k'}^{-\frac{1}{2}} \Sigma_k^{\frac{1}{2}} \left( x - \Sigma_k^{-\frac{1}{2}}\mu_{k'} \right) \right| \geq \frac{|x - \Sigma_k^{-\frac{1}{2}}\mu_{k'}|}{\left\| \left( \Sigma_{k'}^{-\frac{1}{2}} \Sigma_k^{\frac{1}{2}} \right)^{-1} \right\|_{\mathbf{op}}} = \frac{|x - \Sigma_k^{-\frac{1}{2}}\mu_{k'}|}{\sigma},$$

then $|x - \Sigma_k^{-\frac{1}{2}}\mu_{k'}| < \alpha_{k,k'}\sigma$, where $\sigma = \|\Sigma_k^{-\frac{1}{2}}\Sigma_{k'}^{\frac{1}{2}}\|_{\mathrm{op}}$. On the other hand, from the definition of $\alpha_{k,k'}$, we have

$$\alpha_{k,k'} + \alpha_{k,k'}\sigma = |\Sigma_k^{-\frac{1}{2}}\mu_k - \Sigma_k^{\frac{1}{2}}\mu_{k'}|,$$

and thus $\{x \in \mathbb{R}^m : |x - \Sigma_k^{-\frac{1}{2}}\mu_k| < \alpha_{k,k'}\} \cap \{x \in \mathbb{R}^m : |x - \Sigma_k^{-\frac{1}{2}}\mu_{k'}| < \alpha_{k,k'}\sigma\} = \varnothing$. Therefore we obtain

$$\{x \in \mathbb{R}^m : |x - \Sigma_k^{-\frac{1}{2}}\mu_k| < \alpha_{k,k'}\} \cap \{x \in \mathbb{R}^m : \|x - \Sigma_k^{-\frac{1}{2}}\mu_{k'}\|_{\Sigma_k^{-1}\Sigma_{k'}} < \alpha_{k,k'}\} = \varnothing.$$

Making the change of variables as $y = \Sigma_k^{\frac{1}{2}}x$,

$$\{y \in \mathbb{R}^m : \|y - \mu_{k'}\|_{\Sigma_k} < \alpha_{k,k'}\} \cap \{y \in \mathbb{R}^m : \|y - \mu_{k'}\|_{\Sigma_{k'}} < \alpha_{k,k'}\} = \varnothing.$$

$\qquad\square$

**Theorem A.5** (Theorem 4.2 (IV))**.**

$$\sum_{k=1}^{K}\sum_{k'\neq k}\frac{\pi_k\pi_{k'}}{1-\pi_k}c_{k,k'}\log\left(1+\frac{1-\pi_k}{\pi_k}\frac{|\Sigma_k|^{\frac{1}{2}}}{\max_l|\Sigma_l|^{\frac{1}{2}}}\exp\left(-\frac{\left(1+\|\Sigma_{k'}^{-\frac{1}{2}}\Sigma_k^{\frac{1}{2}}\|_{\mathrm{op}}\right)^2}{2}\alpha_{k',k}^2\right)\right)$$

$$\leq\left|H[q]-\widetilde{H}[q]\right|,$$

*where the coefficient $c_{k,k'}$ is defined by*

$$c_{k,k'}:=\frac{1}{\sqrt{(2\pi)^m}}\int_{\mathbb{R}^m_{k,k'}}\exp\left(-\frac{|y|}{2}\right)dy\geq 0,$$

*and the set $\mathbb{R}^m_{k,k'}$ is defined by*

$$\mathbb{R}^m_{k,k'}:=\left\{y\in\mathbb{R}^m:\ \begin{array}{l}y\cdot y\geq(\Sigma_k^{\frac{1}{2}}\Sigma_{k'}^{-1}\Sigma_k^{\frac{1}{2}}y)\cdot y,\\ y\cdot(\Sigma_k^{\frac{1}{2}}\Sigma_{k'}^{-1}(\mu_{k'}-\mu_k))\geq 0\end{array}\right\}. \tag{A.5}$$

*Proof.* Using the equality in (A.1), we write

$$\left|H[q]-\widetilde{H}[q]\right|$$

$$=\sum_{k=1}^{K}\pi_k\int_{\mathbb{R}^m}\frac{1}{\sqrt{(2\pi)^m}}\exp\left(-\frac{|y|^2}{2}\right)$$

$$\times\log\left(1+\sum_{k'\neq k}\frac{\pi_{k'}|\Sigma_k|^{\frac{1}{2}}}{\pi_k|\Sigma_{k'}|^{\frac{1}{2}}}\exp\left(\frac{|y|^2-\left\|\Sigma_k^{\frac{1}{2}}\left(y-\Sigma_k^{-\frac{1}{2}}(\mu_{k'}-\mu_k)\right)\right\|_{\Sigma_{k'}}^2}{2}\right)\right)dy$$

$$\geq\sum_{k=1}^{K}\pi_k\int_{\mathbb{R}^m}\frac{1}{\sqrt{(2\pi)^m}}\exp\left(-\frac{|y|^2}{2}\right)$$

$$\times\log\left(1+\frac{1-\pi_k}{\pi_k}\frac{|\Sigma_k|^{\frac{1}{2}}}{\max_l|\Sigma_l|^{\frac{1}{2}}}\sum_{k'\neq k}\frac{\pi_{k'}}{1-\pi_k}\exp\left(\frac{|y|^2-\left\|\Sigma_k^{\frac{1}{2}}\left(y-\Sigma_k^{-\frac{1}{2}}(\mu_{k'}-\mu_k)\right)\right\|_{\Sigma_{k'}}^2}{2}\right)\right)dy.$$

Since $\log(1+\lambda x)$ is a concave function of $x>0$ for $\lambda>0$, we estimate from below

$$\left|H[q]-\widetilde{H}[q]\right|$$

$$\geq\sum_{k=1}^{K}\pi_k\int_{\mathbb{R}^m}\frac{1}{\sqrt{(2\pi)^m}}\exp\left(-\frac{|y|^2}{2}\right)\sum_{k'\neq k}\frac{\pi_{k'}}{1-\pi_k}$$

$$\times\log\left(1+\frac{1-\pi_k}{\pi_k}\frac{|\Sigma_k|^{\frac{1}{2}}}{\max_l|\Sigma_l|^{\frac{1}{2}}}\exp\left(\frac{|y|^2-\left\|\Sigma_k^{\frac{1}{2}}\left(y-\Sigma_k^{-\frac{1}{2}}(\mu_{k'}-\mu_k)\right)\right\|_{\Sigma_{k'}}^2}{2}\right)\right)dy$$

$$\geq\sum_{k=1}^{K}\pi_k\int_{\mathbb{R}^m_{k,k'}}\frac{1}{\sqrt{(2\pi)^m}}\exp\left(-\frac{|y|^2}{2}\right)\sum_{k'\neq k}\frac{\pi_{k'}}{1-\pi_k}$$

$$\times\log\left(1+\frac{1-\pi_k}{\pi_k}\frac{|\Sigma_k|^{\frac{1}{2}}}{\max_l|\Sigma_l|^{\frac{1}{2}}}\exp\left(\frac{|y|^2-\left\|\Sigma_k^{\frac{1}{2}}\left(y-\Sigma_k^{-\frac{1}{2}}(\mu_{k'}-\mu_k)\right)\right\|_{\Sigma_{k'}}^2}{2}\right)\right)dy.$$

Here, it follows from the two conditions in the definition (A.5) of $\mathbb{R}^m_{k,k'}$ that

$$
|y|^2 - \left\| \Sigma_k^{\frac{1}{2}} \left( y - \Sigma_k^{-\frac{1}{2}} (\mu_{k'} - \mu_k) \right) \right\|^2_{\Sigma_{k'}} \geq \left| \Sigma_{k'}^{-\frac{1}{2}} \Sigma_k^{\frac{1}{2}} y \right|^2 - \left| \Sigma_{k'}^{-\frac{1}{2}} \Sigma_k^{\frac{1}{2}} y - \Sigma_{k'}^{-\frac{1}{2}} (\mu_{k'} - \mu_k) \right|^2
$$

$$
= - \left| \Sigma_{k'}^{-\frac{1}{2}} (\mu_{k'} - \mu_k) \right|^2 + 2 y \cdot \left( \Sigma_k^{\frac{1}{2}} \Sigma_{k'}^{-1} (\mu_{k'} - \mu_k) \right)
$$

$$
\geq - \left| \Sigma_{k'}^{-\frac{1}{2}} (\mu_{k'} - \mu_k) \right|^2
$$

$$
= - \left( 1 + \| \Sigma_{k'}^{-\frac{1}{2}} \Sigma_k^{\frac{1}{2}} \|_{\mathrm{op}} \right)^2 \alpha_{k',k}^2,
$$

for $y \in \mathbb{R}^m_{k,k'}$, where we used the cosine formula in the second step. Combining the above estimates, we conclude that

$$
\left| H[q] - \widetilde{H}[q] \right|
$$

$$
\geq \sum_{k=1}^K \pi_k \int_{\mathbb{R}^m_{k,k'}} \frac{1}{\sqrt{(2\pi)^m}} \exp\left( -\frac{|y|^2}{2} \right) \sum_{k' \neq k} \frac{\pi_{k'}}{1 - \pi_k}
$$

$$
\times \log \left( 1 + \frac{1 - \pi_k}{\pi_k} \frac{|\Sigma_k|^{\frac{1}{2}}}{\max_l |\Sigma_l|^{\frac{1}{2}}} \exp\left( - \frac{\left( 1 + \| \Sigma_{k'}^{-\frac{1}{2}} \Sigma_k^{\frac{1}{2}} \|_{\mathrm{op}} \right)^2}{2} \alpha_{k',k}^2 \right) \right) dy
$$

$$
= \sum_{k=1}^K \sum_{k' \neq k} \frac{\pi_k \pi_{k'}}{1 - \pi_k} c_{k,k'} \log \left( 1 + \frac{1 - \pi_k}{\pi_k} \frac{|\Sigma_k|^{\frac{1}{2}}}{\max_l |\Sigma_l|^{\frac{1}{2}}} \exp\left( - \frac{\left( 1 + \| \Sigma_{k'}^{-\frac{1}{2}} \Sigma_k^{\frac{1}{2}} \|_{\mathrm{op}} \right)^2}{2} \alpha_{k',k}^2 \right) \right).
$$

$\square$

**Remark A.6.** *Either $c_{k,k'}$ or $c_{k',k}$ is positive. Indeed,*

- *if $\Sigma_k^{-1} - \Sigma_{k'}^{-1}$ has at least one positive eigenvalue, then $c_{k,k'}$ is positive;*

- *if all eigenvalues of $\Sigma_k^{-1} - \Sigma_{k'}^{-1} \neq O$ are non-positive, then $\Sigma_{k'}^{-1} - \Sigma_k^{-1}$ has at least one positive eigenvalue;*

- *if $\Sigma_k^{-1} - \Sigma_{k'}^{-1} = O$, then $c_{k,k'} = 1/2$ because $\mathbb{R}^m_{k,k'}$ is a half space of $\mathbb{R}^m$.*

*Here, $O$ is the zero matrix.*

## A.2 Proof of Corollary 4.3

**Corollary A.7** (Corollary 4.3). *Assume $\{\mu_k\}_k$ and $\{\Sigma_k\}_k$ such that*

$$
\frac{\Sigma_k^{-\frac{1}{2}} (\mu_k - \mu_{k'})}{1 + \| \Sigma_k^{-\frac{1}{2}} \Sigma_{k'}^{\frac{1}{2}} \|_{\mathrm{op}}} \sim \mathcal{N}(0, c^2 I), \tag{A.6}
$$

*for all pairs $k, k' \in [K]$ ($k \neq k'$). Then, for $\varepsilon > 0$ and $s \in (0,1)$,*

$$
P\left( \left| H[q] - \widetilde{H}[q] \right| \geq \varepsilon \right) \leq \frac{2(K-1)}{\varepsilon} \left( \sqrt{1-s} \left( 1 + \frac{sc^2}{2} \right) \right)^{-\frac{m}{2}}.
$$

*Proof.* Using Theorem A.2 and Markov's inequality, for $\varepsilon > 0$ and $s \in (0, 1)$, we estimate

$$P\left(\left|H[q] - \widetilde{H}[q]\right| \geq \varepsilon\right) \leq \frac{E\left[\left|H[q] - \widetilde{H}[q]\right|\right]}{\varepsilon} \leq \frac{2}{\varepsilon(1-s)^{\frac{m}{4}}} \sum_{k=1}^{K} \sum_{k' \neq k} \sqrt{\pi_k \pi_{k'}} E\left[\exp\left(-\frac{s\alpha_{k,k'}^2}{4}\right)\right].$$

By the assumption (A.6), $\alpha_{k,k'}^2 / c^2$ follows the $\chi^2$-distribution with $m$ degrees of freedom, that is,

$$\frac{1}{c^2}\left|\frac{\Sigma_k^{-\frac{1}{2}}(\mu_k - \mu_{k'})}{1 + \|\Sigma_k^{-\frac{1}{2}}\Sigma_{k'}^{\frac{1}{2}}\|_{\mathrm{op}}}\right|^2 \sim \chi_m^2.$$

Therefore, we conclude from the moment generating function for $\chi^2$-distribution that

$$\begin{aligned}
P\left(\left|H[q] - \widetilde{H}[q]\right| \geq \varepsilon\right) &\leq \frac{2}{\varepsilon(1-s)^{\frac{m}{4}}} \sum_{k=1}^{K} \sum_{k' \neq k} \sqrt{\pi_k \pi_{k'}} E\left[\exp\left(-\frac{sc^2}{4}\frac{\alpha_{k,k'}^2}{c^2}\right)\right] \\
&= \frac{2}{\varepsilon(1-s)^{\frac{m}{4}}} \underbrace{\sum_{k=1}^{K} \sum_{k' \neq k} \sqrt{\pi_k \pi_{k'}}}_{\leq \sum_{k=1}^{K}\sum_{k' \neq k}\frac{\pi_k + \pi_{k'}}{2} = K-1} \left(1 - 2\left(-\frac{sc^2}{4}\right)\right)^{-\frac{m}{2}} \\
&\leq \frac{2(K-1)}{\varepsilon} \left(\sqrt{1-s}\left(1 + \frac{sc^2}{2}\right)\right)^{-\frac{m}{2}}.
\end{aligned}$$

$\square$

### A.3 Proof of Theorem 4.4

**Theorem A.8** (Theorem 4.4). *For $k \in [K]$, $p, q \in [m]$, and $s \in (0, 1)$,*

(i) $\left|\dfrac{\partial}{\partial \mu_{k,p}}\left(H[q] - \widetilde{H}[q]\right)\right|$

$$\leq \frac{2}{(1-s)^{\frac{m+2}{4}}} \sum_{k' \neq k} \sqrt{\pi_k \pi_{k'}}\left(\left\|\Gamma_{k'}^{-1}\right\|_1 + \left\|\Gamma_k^{-1}\right\|_1\right)\exp\left(-\frac{s\max(\alpha_{k,k'}, \alpha_{k',k})^2}{4}\right), \tag{A.7}$$

(ii) $\left|\dfrac{\partial}{\partial \gamma_{k,pq}}\left(H[q] - \widetilde{H}[q]\right)\right|$

$$\leq \frac{6}{(1-s)^{\frac{m+4}{4}}} \sum_{k' \neq k} \sqrt{\pi_k \pi_{k'}}\left(2|\Gamma_k|^{-1}|\Gamma_{k,pq}| + \left\|\Gamma_k^{-1}\right\|_1 + \left\|\Gamma_{k'}^{-1}\right\|_1\right)\exp\left(-\frac{s\max(\alpha_{k,k'}, \alpha_{k',k})^2}{4}\right), \tag{A.8}$$

(iii) $\left|\dfrac{\partial}{\partial \pi_k}\left(H[q] - \widetilde{H}[q]\right)\right| \leq \dfrac{8}{(1-s)^{\frac{m}{4}}} \displaystyle\sum_{k' \neq k} \sqrt{\dfrac{\pi_{k'}}{\pi_k}}\exp\left(-\dfrac{s\max(\alpha_{k,k'}, \alpha_{k',k})^2}{4}\right), \tag{A.9}$

*where $\mu_{k,p}$ and $\gamma_{k,pq}$ is the $p$-th and $(p,q)$-th components of vector $\mu_k$ and matrix $\Gamma_k$, respectively, and $\|\cdot\|_1$ is the entry-wise matrix 1-norm, and $|\Gamma_{k,pq}|$ is the determinant of the $(m-1) \times (m-1)$ matrix that results from deleting $p$-th row and $q$-th column of matrix $\Gamma_k$.*

*Proof.* Using the equality (A.1), we have

$$
\begin{aligned}
H[q] - \widetilde{H}[q] = \sum_{k=1}^{K} \pi_k & \int_{\mathbb{R}^m} \frac{1}{\sqrt{(2\pi)^m}} \exp\left(-\frac{|y|^2}{2}\right) \\
& \times \log\left(1 + \sum_{k' \neq k} \frac{\pi_{k'}|\Gamma_k|}{\pi_k|\Gamma_{k'}|} \exp\left(\frac{|y|^2 - \left|\Gamma_{k'}^{-1}\Gamma_k y - \Gamma_{k'}^{-1}(\mu_{k'} - \mu_k)\right|^2}{2}\right)\right) dy \\
= \pi_k & \int_{\mathbb{R}^m} \frac{1}{\sqrt{(2\pi)^m}} \exp\left(-\frac{|y|^2}{2}\right) \\
& \times \log\left(1 + \sum_{k' \neq k} \frac{\pi_{k'}|\Gamma_k|}{\pi_k|\Gamma_{k'}|} \exp\left(\frac{|y|^2 - \left|\Gamma_{k'}^{-1}\Gamma_k y - \Gamma_{k'}^{-1}(\mu_{k'} - \mu_k)\right|^2}{2}\right)\right) dy \\
+ \sum_{\ell \neq k} \pi_\ell & \int_{\mathbb{R}^m} \frac{1}{\sqrt{(2\pi)^m}} \exp\left(-\frac{|y|^2}{2}\right) \\
& \times \log\left(1 + \underbrace{\sum_{k' \neq \ell} \frac{\pi_{k'}|\Gamma_\ell|}{\pi_\ell|\Gamma_{k'}|} \exp\left(\frac{|y|^2 - \left|\Gamma_{k'}^{-1}\Gamma_\ell y - \Gamma_{k'}^{-1}(\mu_{k'} - \mu_\ell)\right|^2}{2}\right)}_{= \frac{\pi_k|\Gamma_\ell|}{\pi_\ell|\Gamma_k|} \exp\left(\frac{|y|^2 - \left|\Gamma_k^{-1}\Gamma_\ell y - \Gamma_k^{-1}(\mu_k - \mu_\ell)\right|^2}{2}\right) + \text{terms independent of } k}\right) dy.
\end{aligned}
$$

(i)

$$
\begin{aligned}
& \frac{\partial}{\partial \mu_{k,p}} \left(H[q] - \widetilde{H}[q]\right) \\
& = \pi_k \int_{\mathbb{R}^m} \frac{1}{\sqrt{(2\pi)^m}} \exp\left(-\frac{|y|^2}{2}\right) \\
& \quad \times \frac{\sum_{k' \neq k} \frac{\pi_{k'}|\Gamma_k|}{\pi_k|\Gamma_{k'}|} \exp\left(\frac{|y|^2 - \left|\Gamma_{k'}^{-1}\Gamma_k y - \Gamma_{k'}^{-1}(\mu_{k'} - \mu_k)\right|^2}{2}\right) \frac{\partial}{\partial \mu_{k,p}}\left(\frac{-\left|\Gamma_{k'}^{-1}\Gamma_k y - \Gamma_{k'}^{-1}(\mu_{k'} - \mu_k)\right|^2}{2}\right)}{1 + \sum_{k' \neq k} \frac{\pi_{k'}|\Gamma_k|}{\pi_k|\Gamma_{k'}|} \exp\left(\frac{|y|^2 - \left|\Gamma_{k'}^{-1}\Gamma_k y - \Gamma_{k'}^{-1}(\mu_{k'} - \mu_k)\right|^2}{2}\right)} dy \\
& \quad + \sum_{\ell \neq k} \pi_\ell \int_{\mathbb{R}^m} \frac{1}{\sqrt{(2\pi)^m}} \exp\left(-\frac{|y|^2}{2}\right) \\
& \quad \times \frac{\frac{\pi_k|\Gamma_\ell|}{\pi_\ell|\Gamma_k|} \exp\left(\frac{|y|^2 - \left|\Gamma_k^{-1}\Gamma_\ell y - \Gamma_k^{-1}(\mu_k - \mu_\ell)\right|^2}{2}\right) \frac{\partial}{\partial \mu_{k,p}}\left(\frac{-\left|\Gamma_k^{-1}\Gamma_\ell y - \Gamma_k^{-1}(\mu_k - \mu_\ell)\right|^2}{2}\right)}{1 + \sum_{k' \neq \ell} \frac{\pi_{k'}|\Gamma_\ell|}{\pi_\ell|\Gamma_{k'}|} \exp\left(\frac{|y|^2 - \left|\Gamma_{k'}^{-1}\Gamma_\ell y - \Gamma_{k'}^{-1}(\mu_{k'} - \mu_\ell)\right|^2}{2}\right)} dy.
\end{aligned} \tag{A.10}
$$

Since $\frac{x}{a+x} \leq x^{\frac{1}{2}}$ for $x > 1$ when $a \geq 1$, we estimate

$$
\begin{aligned}
& \frac{\frac{\pi_{k'}|\Gamma_k|}{\pi_k|\Gamma_{k'}|} \exp\left(\frac{|y|^2 - \left|\Gamma_{k'}^{-1}\Gamma_k y - \Gamma_{k'}^{-1}(\mu_{k'} - \mu_k)\right|^2}{2}\right)}{1 + \sum_{k' \neq k} \frac{\pi_{k'}|\Gamma_k|}{\pi_k|\Gamma_{k'}|} \exp\left(\frac{|y|^2 - \left|\Gamma_{k'}^{-1}\Gamma_k y - \Gamma_{k'}^{-1}(\mu_{k'} - \mu_k)\right|^2}{2}\right)} \\
& \qquad\qquad \leq \sqrt{\frac{\pi_{k'}|\Gamma_k|}{\pi_k|\Gamma_{k'}|}} \exp\left(\frac{|y|^2 - \left|\Gamma_{k'}^{-1}\Gamma_k y - \Gamma_{k'}^{-1}(\mu_{k'} - \mu_k)\right|^2}{4}\right),
\end{aligned} \tag{A.11}
$$

$$\frac{\frac{\pi_k|\Gamma_\ell|}{\pi_\ell|\Gamma_k|} \exp\left(\frac{|y|^2 - \left|\Gamma_k^{-1}\Gamma_\ell y - \Gamma_k^{-1}(\mu_k - \mu_\ell)\right|^2}{2}\right)}{1 + \sum_{k'\neq\ell} \frac{\pi_{k'}|\Gamma_\ell|}{\pi_\ell|\Gamma_{k'}|} \exp\left(\frac{|y|^2 - \left|\Gamma_{k'}^{-1}\Gamma_\ell y - \Gamma_{k'}^{-1}(\mu_{k'} - \mu_\ell)\right|^2}{2}\right)} \tag{A.12}$$

$$\leq \sqrt{\frac{\pi_k|\Gamma_\ell|}{\pi_\ell|\Gamma_k|}} \exp\left(\frac{|y|^2 - \left|\Gamma_k^{-1}\Gamma_\ell y - \Gamma_k^{-1}(\mu_k - \mu_\ell)\right|^2}{4}\right).$$

We also calculate

$$\frac{\partial}{\partial\mu_{k,p}} \left(\frac{-\left|\Gamma_{k'}^{-1}\Gamma_k y - \Gamma_{k'}^{-1}(\mu_{k'} - \mu_k)\right|^2}{2}\right) = \sum_{i=1}^m \left[\Gamma_{k'}^{-1}\Gamma_k y - \Gamma_{k'}^{-1}(\mu_{k'} - \mu_k)\right]_i \gamma_{k',ip}^{-1}, \tag{A.13}$$

and

$$\frac{\partial}{\partial\mu_{k,p}} \left(\frac{-\left|\Gamma_k^{-1}\Gamma_\ell y - \Gamma_k^{-1}(\mu_k - \mu_\ell)\right|^2}{2}\right) = \sum_{i=1}^m \left[\Gamma_k^{-1}\Gamma_\ell y - \Gamma_k^{-1}(\mu_k - \mu_\ell)\right]_i \gamma_{k,ip}^{-1}, \tag{A.14}$$

where we denote by $[v]_i$ the $i$-th component of vector $v$, and $\gamma_{k',ip}^{-1}$ and $\gamma_{k,ip}^{-1}$ the $(i,p)$-th component of matrix $\Gamma_{k'}^{-1}$ and $\Gamma_k^{-1}$, respectively. By using (A.10)–(A.14),

$$\left|\frac{\partial}{\partial\mu_{k,p}}\left(H[q] - \widetilde{H}[q]\right)\right|$$

$$\leq \sum_{k'\neq k} \sqrt{\pi_k\pi_{k'}} \sum_{i=1}^m \left|\gamma_{k',ip}^{-1}\right|$$

$$\times \underbrace{\sqrt{\frac{|\Gamma_k|}{|\Gamma_{k'}|}} \int_{\mathbb{R}^m} \frac{\left[\Gamma_{k'}^{-1}\Gamma_k y - \Gamma_{k'}^{-1}(\mu_{k'} - \mu_k)\right]_i}{\sqrt{(2\pi)^m}} \exp\left(\frac{-|y|^2 - \left|\Gamma_{k'}^{-1}\Gamma_k y - \Gamma_{k'}^{-1}(\mu_{k'} - \mu_k)\right|^2}{4}\right) dy}_{\leq \frac{2}{(1-s)^{\frac{m+2}{4}}} \exp\left(-\frac{s\alpha_{k,k'}^2}{4}\right)}$$

$$+ \sum_{\ell\neq k} \sqrt{\pi_k\pi_\ell} \sum_{i=1}^m \left|\gamma_{k,ip}^{-1}\right|$$

$$\times \underbrace{\sqrt{\frac{|\Gamma_\ell|}{|\Gamma_k|}} \int_{\mathbb{R}^m} \frac{\left[\Gamma_k^{-1}\Gamma_\ell y - \Gamma_k^{-1}(\mu_k - \mu_\ell)\right]_i}{\sqrt{(2\pi)^m}} \exp\left(\frac{-|y|^2 - \left|\Gamma_k^{-1}\Gamma_\ell y - \Gamma_k^{-1}(\mu_k - \mu_\ell)\right|^2}{4}\right) dy}_{\leq \frac{2}{(1-s)^{\frac{m+2}{4}}} \exp\left(-\frac{s\alpha_{\ell,k}^2}{4}\right)}$$

$$\leq \frac{2}{(1-s)^{\frac{m+2}{4}}} \sum_{k'\neq k} \sqrt{\pi_k\pi_{k'}} \left\{\underbrace{\left(\sum_{i=1}^m \left|\gamma_{k',ip}^{-1}\right|\right)}_{\leq\left\|\Gamma_{k'}^{-1}\right\|_1} \exp\left(-\frac{s\alpha_{k,k'}^2}{4}\right) + \underbrace{\left(\sum_{i=1}^m \left|\gamma_{k,ip}^{-1}\right|\right)}_{\leq\left\|\Gamma_k^{-1}\right\|_1} \exp\left(-\frac{s\alpha_{k',k}^2}{4}\right)\right\},$$

where the last inequality is given by the same arguments in the proof of Theorem A.2. Similarly Theorem A.3, this evaluation holds when $\alpha_{k,k'}$ and $\alpha_{k',k}$ are replaced with any $\alpha \leq \alpha_{\{k,k'\}}$, for example $\max(\alpha_{k,k'}, \alpha_{k',k})$.

**(ii)**

$$\frac{\partial}{\partial \gamma_{k,pq}} \left( H[q] - \widetilde{H}[q] \right)$$

$$= \pi_k \int_{\mathbb{R}^m} \frac{1}{\sqrt{(2\pi)^m}} \exp\left( -\frac{|y|^2}{2} \right) \frac{\sum_{k' \neq k} \frac{\pi_{k'} |\Gamma_k|}{\pi_k |\Gamma_{k'}|} \exp\left( \frac{|y|^2 - \left|\Gamma_{k'}^{-1} \Gamma_k y - \Gamma_{k'}^{-1} (\mu_{k'} - \mu_k)\right|^2}{2} \right)}{1 + \sum_{k' \neq k} \frac{\pi_{k'} |\Gamma_k|}{\pi_k |\Gamma_{k'}|} \exp\left( \frac{|y|^2 - \left|\Gamma_{k'}^{-1} \Gamma_k y - \Gamma_{k'}^{-1} (\mu_{k'} - \mu_k)\right|^2}{2} \right)}$$

$$\times \left\{ \underbrace{|\Gamma_k|^{-1} \left( \frac{\partial}{\partial \gamma_{k,pq}} |\Gamma_k| \right)}_{= |\Gamma_k|^{-1} |\Gamma_{k,pq}|} + \frac{\partial}{\partial \gamma_{k,pq}} \left( \frac{-\left|\Gamma_{k'}^{-1} \Gamma_k y - \Gamma_{k'}^{-1} (\mu_{k'} - \mu_k)\right|^2}{2} \right) \right\} dy$$

$$+ \sum_{\ell \neq k} \pi_\ell \int_{\mathbb{R}^m} \frac{1}{\sqrt{(2\pi)^m}} \exp\left( -\frac{|y|^2}{2} \right) \frac{\frac{\pi_k |\Gamma_\ell|}{\pi_\ell |\Gamma_k|} \exp\left( \frac{|y|^2 - \left|\Gamma_k^{-1} \Gamma_\ell y - \Gamma_k^{-1} (\mu_k - \mu_\ell)\right|^2}{2} \right)}{1 + \sum_{k' \neq \ell} \frac{\pi_{k'} |\Gamma_\ell|}{\pi_\ell |\Gamma_{k'}|} \exp\left( \frac{|y|^2 - \left|\Gamma_{k'}^{-1} \Gamma_\ell y - \Gamma_{k'}^{-1} (\mu_{k'} - \mu_\ell)\right|^2}{2} \right)}$$

$$\times \left\{ \underbrace{|\Gamma_k| \frac{\partial}{\partial \gamma_{k,pq}} |\Gamma_k|^{-1}}_{= |\Gamma_k|^{-1} |\Gamma_{k,pq}|} + \frac{\partial}{\partial \gamma_{k,pq}} \left( \frac{-\left|\Gamma_k^{-1} \Gamma_\ell y - \Gamma_k^{-1} (\mu_k - \mu_\ell)\right|^2}{2} \right) \right\} dy. \tag{A.15}$$

We calculate

$$\frac{\partial}{\partial \gamma_{k,pq}} \left( \frac{-\left|\Gamma_{k'}^{-1} \Gamma_k y - \Gamma_{k'}^{-1} (\mu_{k'} - \mu_k)\right|^2}{2} \right) = -\sum_{i=1}^m \left[ \Gamma_{k'}^{-1} \Gamma_k y - \Gamma_{k'}^{-1} (\mu_{k'} - \mu_k) \right]_i \gamma_{k',ip}^{-1} y_q, \tag{A.16}$$

and

$$\frac{\partial}{\partial \gamma_{k,pq}} \left( \frac{-\left|\Gamma_k^{-1} \Gamma_\ell y - \Gamma_k^{-1} (\mu_k - \mu_\ell)\right|^2}{2} \right)$$

$$= -\sum_{i=1}^m \left[ \Gamma_k^{-1} \Gamma_\ell y - \Gamma_k^{-1} (\mu_k - \mu_\ell) \right]_i \left[ \left( \frac{\partial}{\partial \gamma_{k,pq}} \Gamma_k^{-1} \right) \Gamma_k \left( \Gamma_k^{-1} \Gamma_\ell y - \Gamma_k^{-1} (\mu_k - \mu_\ell) \right) \right]_i, \tag{A.17}$$

where we denote by $[v]_i$ the $i$-th component of vector $v$, and $\gamma_{k',ip}^{-1}$ and $\gamma_{k,ip}^{-1}$ the $(i,p)$-th component of matrix $\Gamma_{k'}^{-1}$ and $\Gamma_k^{-1}$, respectively, and $y_q$ is the $q$-th component of vector $y$, and $\frac{\partial}{\partial \gamma_{k,pq}} \Gamma_k^{-1}$ is component-wise derivative of matrix $\Gamma_k^{-1}$ with respect to $\gamma_{k,pq}$. We also calulate

$$\left( \frac{\partial}{\partial \gamma_{k,pq}} \Gamma_k^{-1} \right) \Gamma_k = \delta_{k,pq} \Gamma_k^{-1},$$

where $\delta_{k,pq}$ is the matrix such that $(p,q)$-th component is one, and other are zero. We further estimate (A.17) by

$$\left| \frac{\partial}{\partial \gamma_{k,pq}} \left( \frac{-\left|\Gamma_k^{-1} \Gamma_\ell y - \Gamma_k^{-1} (\mu_k - \mu_\ell)\right|^2}{2} \right) \right|$$

$$= \left| \left\langle \Gamma_k^{-1} \Gamma_\ell y - \Gamma_k^{-1} (\mu_k - \mu_\ell), \delta_{k,pq} \Gamma_k^{-1} \left( \Gamma_k^{-1} \Gamma_\ell y - \Gamma_k^{-1} (\mu_k - \mu_\ell) \right) \right\rangle \right| \tag{A.18}$$

$$\leq \sum_{i=1}^m \sum_{j=1}^m \left| \left[ \delta_{k,pq} \Gamma_k^{-1} \right]_{ij} \right| \left| \left[ \Gamma_k^{-1} \Gamma_\ell y - \Gamma_k^{-1} (\mu_k - \mu_\ell) \right]_i \left[ \Gamma_k^{-1} \Gamma_\ell y - \Gamma_k^{-1} (\mu_k - \mu_\ell) \right]_j \right|,$$

where $\left[ \delta_{k,pq} \Gamma_k^{-1} \right]_{ij}$ is $(i,j)$-th component of matrix $\delta_{k,pq} \Gamma_k^{-1}$.

By using inequality of $\frac{x}{a+x} \le x^{\frac{1}{2}}$ for $x > 1$ $(a \ge 1)$, and the arguments (A.15)–(A.18), we have

$$
\left| \frac{\partial}{\partial \gamma_{k,pq}} \left( H[q] - \widetilde{H}[q] \right) \right|
$$

$$
\le \sum_{k' \ne k} \sqrt{\pi_k \pi_{k'}} |\Gamma_k|^{-1} |\Gamma_{k,pq}|
$$

$$
\times \underbrace{\sqrt{\frac{|\Gamma_k|}{|\Gamma_{k'}|}} \int_{\mathbb{R}^m} \frac{1}{\sqrt{(2\pi)^m}} \exp \left( \frac{-|y|^2 - \left| \Gamma_{k'}^{-1} \Gamma_k y - \Gamma_{k'}^{-1} (\mu_{k'} - \mu_k) \right|^2}{4} \right) dy}_{\le \frac{2}{(1-s)^{\frac{m}{4}}} \exp \left( -\frac{s\alpha_{k,k'}^2}{4} \right)}
$$

$$
+ \sum_{k' \ne k} \sqrt{\pi_k \pi_{k'}} \sum_{i=1}^{m} \left| \gamma_{k',ip}^{-1} \right|
$$

$$
\times \underbrace{\sqrt{\frac{|\Gamma_k|}{|\Gamma_{k'}|}} \int_{\mathbb{R}^m} \frac{\left[ \Gamma_{k'}^{-1} \Gamma_k y - \Gamma_{k'}^{-1} (\mu_{k'} - \mu_k) \right]_i}{\sqrt{(2\pi)^m}} \exp \left( \frac{-|y|^2 - \left| \Gamma_{k'}^{-1} \Gamma_k y - \Gamma_{k'}^{-1} (\mu_{k'} - \mu_k) \right|^2}{4} \right) dy}_{\le \frac{2}{(1-s)^{\frac{m+2}{4}}} \exp \left( -\frac{s\alpha_{k,k'}^2}{4} \right)}
$$

$$
+ \sum_{\ell \ne k} \sqrt{\pi_k \pi_\ell} \sum_{i=1}^{m} |\Gamma_k|^{-1} |\Gamma_{k,pq}|
$$

$$
\times \underbrace{\sqrt{\frac{|\Gamma_\ell|}{|\Gamma_k|}} \int_{\mathbb{R}^m} \frac{1}{\sqrt{(2\pi)^m}} \exp \left( \frac{-|y|^2 - \left| \Gamma_k^{-1} \Gamma_\ell y - \Gamma_k^{-1} (\mu_k - \mu_\ell) \right|^2}{4} \right) dy}_{\le \frac{2}{(1-s)^{\frac{m}{4}}} \exp \left( -\frac{s\alpha_{\ell,k}^2}{4} \right)}
$$

$$
+ \sum_{\ell \ne k} \sqrt{\pi_k \pi_\ell} \sum_{i=1}^{m} \sum_{j=1}^{m} \left| \left[ \delta_{k,pq} \Gamma_k^{-1} \right]_{ij} \right|
$$

$$
\times \underbrace{\sqrt{\frac{|\Gamma_\ell|}{|\Gamma_k|}} \int_{\mathbb{R}^m} \frac{\left| \left[ \Gamma_k^{-1} \Gamma_\ell y - \Gamma_k^{-1} (\mu_k - \mu_\ell) \right]_i \left[ \Gamma_k^{-1} \Gamma_\ell y - \Gamma_k^{-1} (\mu_k - \mu_\ell) \right]_j \right|}{\sqrt{(2\pi)^m}}}_{}
$$

$$
\underbrace{\times \exp \left( \frac{-|y|^2 - \left| \Gamma_k^{-1} \Gamma_\ell y - \Gamma_k^{-1} (\mu_k - \mu_\ell) \right|^2}{4} \right) dy}_{\le \frac{6}{(1-s)^{\frac{m+4}{4}}} \exp \left( -\frac{s\alpha_{\ell,k}^2}{4} \right)}
$$

$$
\le \frac{6}{(1-s)^{\frac{m+4}{4}}} \sum_{k' \ne k} \sqrt{\pi_k \pi_{k'}} \Bigg[ \left\{ |\Gamma_k|^{-1} |\Gamma_{k,pq}| + \underbrace{\left( \sum_{i=1}^{m} \left| \gamma_{k',ip}^{-1} \right| \right)}_{\le \|\Gamma_{k'}^{-1}\|_1} \right\} \exp \left( -\frac{s\alpha_{k,k'}^2}{4} \right)
$$

$$
+ \left\{ |\Gamma_k|^{-1} |\Gamma_{k,pq}| + \underbrace{\left( \sum_{i=1}^{m} \sum_{j=1}^{m} \left| \left[ \delta_{k,pq} \Gamma_k^{-1} \right]_{ij} \right| \right)}_{\le \|\Gamma_k^{-1}\|_1} \right\} \exp \left( -\frac{s\alpha_{k',k}^2}{4} \right) \Bigg],
$$

where the last inequality is given by the same arguments in the proof of Theorem A.2.

**(iii)**

$$
\frac{\partial}{\partial \pi_k} \left( H[q] - \widetilde{H}[q] \right)
$$

$$
= \int_{\mathbb{R}^m} \frac{1}{\sqrt{(2\pi)^m}} \exp\left( -\frac{|y|^2}{2} \right)
$$

$$
\times \log \left( 1 + \sum_{k' \neq k} \frac{\pi_{k'} |\Gamma_k|}{\pi_k |\Gamma_{k'}|} \exp\left( \frac{|y|^2 - \left| \Gamma_{k'}^{-1} \Gamma_k y - \Gamma_{k'}^{-1} (\mu_{k'} - \mu_k) \right|^2}{2} \right) \right) dy
$$

$$
+ \pi_k \int_{\mathbb{R}^m} \frac{1}{\sqrt{(2\pi)^m}} \exp\left( -\frac{|y|^2}{2} \right)
$$

$$
\times \frac{\sum_{k' \neq k} \frac{-\pi_{k'} |\Gamma_k|}{\pi_k^2 |\Gamma_{k'}|} \exp\left( \frac{|y|^2 - \left| \Gamma_{k'}^{-1} \Gamma_k y - \Gamma_{k'}^{-1} (\mu_{k'} - \mu_k) \right|^2}{2} \right)}{1 + \sum_{k' \neq k} \frac{\pi_{k'} |\Gamma_k|}{\pi_k |\Gamma_{k'}|} \exp\left( \frac{|y|^2 - \left| \Gamma_{k'}^{-1} \Gamma_k y - \Gamma_{k'}^{-1} (\mu_{k'} - \mu_k) \right|^2}{2} \right)} dy \tag{A.19}
$$

$$
+ \sum_{\ell \neq k} \pi_\ell \int_{\mathbb{R}^m} \frac{1}{\sqrt{(2\pi)^m}} \exp\left( -\frac{|y|^2}{2} \right)
$$

$$
\times \frac{\frac{|\Gamma_\ell|}{\pi_\ell |\Gamma_k|} \exp\left( \frac{|y|^2 - \left| \Gamma_k^{-1} \Gamma_\ell y - \Gamma_k^{-1} (\mu_k - \mu_\ell) \right|^2}{2} \right)}{1 + \sum_{k' \neq \ell} \frac{\pi_{k'} |\Gamma_\ell|}{\pi_\ell |\Gamma_{k'}|} \exp\left( \frac{|y|^2 - \left| \Gamma_{k'}^{-1} \Gamma_\ell y - \Gamma_{k'}^{-1} (\mu_{k'} - \mu_\ell) \right|^2}{2} \right)} dy.
$$

Using inequalities of $\log(1+x) \leq \sqrt{x}$ for $x \geq 0$ and $\frac{x}{a+x} \leq x^{\frac{1}{2}}$ for $x > 1$ when $a \geq 1$, we estimate

$$
\left| \frac{\partial}{\partial \pi_k} \left( H[q] - \widetilde{H}[q] \right) \right|
$$

$$
\leq \sum_{k' \neq k} \sqrt{\frac{\pi_{k'}}{\pi_k}} \sqrt{\frac{|\Gamma_k|}{|\Gamma_{k'}|}} \underbrace{\int_{\mathbb{R}^m} \frac{1}{\sqrt{(2\pi)^m}} \exp\left( \frac{-|y|^2 - \left| \Gamma_{k'}^{-1} \Gamma_k y - \Gamma_{k'}^{-1} (\mu_{k'} - \mu_k) \right|^2}{4} \right) dy}_{\leq \frac{2}{(1-s)^{\frac{m}{4}}} \exp\left( -\frac{s\alpha_{k,k'}^2}{4} \right)}
$$

$$
+ \sum_{k' \neq k} \sqrt{\frac{\pi_{k'}}{\pi_k}} \sqrt{\frac{|\Gamma_k|}{|\Gamma_{k'}|}} \underbrace{\int_{\mathbb{R}^m} \frac{1}{\sqrt{(2\pi)^m}} \exp\left( \frac{-|y|^2 - \left| \Gamma_{k'}^{-1} \Gamma_k y - \Gamma_{k'}^{-1} (\mu_{k'} - \mu_k) \right|^2}{4} \right) dy}_{\leq \frac{2}{(1-s)^{\frac{m}{4}}} \exp\left( -\frac{s\alpha_{k,k'}^2}{4} \right)}
$$

$$
+ \sum_{\ell \neq k} \sqrt{\frac{\pi_\ell}{\pi_k}} \sqrt{\frac{|\Gamma_\ell|}{|\Gamma_k|}} \underbrace{\int_{\mathbb{R}^m} \frac{1}{\sqrt{(2\pi)^m}} \exp\left( \frac{-|y|^2 - \left| \Gamma_k^{-1} \Gamma_\ell y - \Gamma_k^{-1} (\mu_k - \mu_\ell) \right|^2}{4} \right) dy}_{\leq \frac{2}{(1-s)^{\frac{m}{4}}} \exp\left( -\frac{s\alpha_{\ell,k}^2}{4} \right)}
$$

$$
\leq \frac{4}{(1-s)^{\frac{m}{4}}} \sum_{k' \neq k} \sqrt{\frac{\pi_{k'}}{\pi_k}} \left\{ \exp\left( -\frac{s\alpha_{k,k'}^2}{4} \right) + \exp\left( -\frac{s\alpha_{k',k}^2}{4} \right) \right\},
$$

where the last inequality is given by the same arguments in the proof of Theorem A.2.

$\square$

# B   Coincident covariance case

In this section, we always assume that the covariance matrices $\Sigma_k$ are coincident for all $k$, that is,

$$\Sigma_k = \Sigma \quad \text{for all } k \in [K], \tag{B.1}$$

where $\Sigma \in \mathbb{R}^{m \times m}$ is a positive definite matrix. In this case, $\alpha_{k,k'}$ has the form

$$\alpha_{k,k'} = \frac{\|\mu_k - \mu_{k'}\|_\Sigma}{2}, \quad k, k' \in [K].$$

## B.1   Proof of Proposition 4.5

**Proposition B.1** (Proposition 4.5). *Let $m \geq K \geq 2$. Then,*

$$H[q] = \widetilde{H}[q] - \sum_{k=1}^{K} \frac{\pi_k}{(2\pi)^{\frac{K-1}{2}}} \int_{\mathbb{R}^{K-1}} \exp\left(-\frac{|v|^2}{2}\right) \log\left(1 + \sum_{k' \neq k} \frac{\pi_{k'}}{\pi_k} \exp\left(\frac{|v|^2 - |v - u_{k',k}|^2}{2}\right)\right) dv, \quad \text{(B.2)}$$

*where $u_{k',k} := [R_k \Sigma^{-1/2}(\mu_{k'} - \mu_k)]_{1:K-1} \in \mathbb{R}^{K-1}$ and $R_k \in \mathbb{R}^{m \times m}$ is some rotation matrix such that*

$$R_k \Sigma^{-\frac{1}{2}}(\mu_{k'} - \mu_k) \in \text{span}\{e_1, \cdots, e_{K-1}\}, \quad k' \in [K]. \tag{B.3}$$

*Here, $\{e_i\}_{i=1}^{K-1}$ is the standard basis in $\mathbb{R}^{K-1}$, and $u_{1:K-1} := (u_1, \ldots, u_{K-1})^T \in \mathbb{R}^{K-1}$ for $u = (u_1, \ldots, u_m)^T \in \mathbb{R}^m$.*

*Proof.* First, we observe that

$$H[q] = \widetilde{H}[q] - \underbrace{\sum_{k=1}^{K} \pi_k \int_{\mathbb{R}^m} \mathcal{N}(x|\mu_k, \Sigma_k) \left\{\log\left(\sum_{k'=1}^{K} \pi_{k'} \mathcal{N}(x|\mu_{k'}, \Sigma_k)\right) - \log\left(\pi_k \mathcal{N}(x|\mu_k, \Sigma_k)\right)\right\} dx}_{=: (\clubsuit)}.$$

Using (A.1) with the assumption (B.1), we write

$$(\clubsuit) = \sum_{k=1}^{K} \pi_k \int_{\mathbb{R}^m} \frac{1}{\sqrt{(2\pi)^m}} \exp\left(-\frac{|y|^2}{2}\right)$$

$$\times \log\left(1 + \sum_{k' \neq k} \frac{\pi_{k'}}{\pi_k} \exp\left(\frac{|y|^2 - \left|\left(y - \Sigma^{-\frac{1}{2}}(\mu_{k'} - \mu_k)\right)\right|^2}{2}\right)\right) dy.$$

We choose the rotation matrix $R_k \in \mathbb{R}^{m \times m}$ satisfying (B.3) for each $k \in [K]$. Making the change of variables as $z = R_k y$, we write

$$(\clubsuit) = \sum_{k=1}^{K} \pi_k \int_{\mathbb{R}^m} \frac{1}{\sqrt{(2\pi)^m}} \exp\left(-\frac{|z|^2}{2}\right)$$

$$\times \log\left(1 + \sum_{k' \neq k} \frac{\pi_{k'}}{\pi_k} \underbrace{\exp\left(\frac{|z|^2 - \left|\left(R^T z - \Sigma^{-\frac{1}{2}}(\mu_{k'} - \mu_k)\right)\right|^2}{2}\right)}_{=\exp\left(\frac{|z|^2 - |z - u_{k',k}|^2}{2}\right)}\right) dy,$$

where $u_{k',k} = [R_k \Sigma^{-1/2}(\mu_{k'} - \mu_k)]_{1:K-1} \in \mathbb{R}^{K-1}$, that is,

$$R_k \Sigma^{-\frac{1}{2}}(\mu_{k'} - \mu_k) = (u_{k',k}, 0, \cdots, 0)^T.$$

We change the variables as

$$z_1 = v_1, \ \ldots, \ z_{K-1} = v_{K-1}, \ z_K = r\cos\theta_K, \ z_{K+1} = r\sin\theta_K\cos\theta_{K+1}, \ \ldots$$

$$z_{m-1} = r\sin\theta_K\cdots\sin\theta_{m-2}\cos\theta_{m-1}, \ z_m = r\sin\theta_K\cdots\sin\theta_{m-2}\sin\theta_{m-1},$$

where $-\infty < v_i < \infty$ $(i = 1, \ldots, K-1)$, $r > 0$, $0 < \theta_j < \pi$ $(j = K, \ldots, m-2)$, and $0 < \theta_{m-1} < 2\pi$. Since

$$|z|^2 = |v|^2 + r^2, \ \ |z - u_{k',k}|^2 = |v - u_{k',k}|^2 + r^2,$$

$$dz_1 \cdots dz_m = r^{m-K} \prod_{i=K}^{m-1} (\sin\theta_i)^{m-i-1} dv_1 \cdots dv_{K-1}\, dr\, d\theta_K \cdots d\theta_{m-1},$$

we obtain

$$
\begin{aligned}
(\clubsuit) &= \sum_{k=1}^{K} \frac{\pi_k}{\sqrt{(2\pi)^m}} \int_0^{2\pi} d\theta_{m-1} \int_0^{\pi} d\theta_{m-2} \cdots \int_0^{\pi} d\theta_K \int_0^{\infty} dr \int_{-\infty}^{\infty} dv_{K-1} \cdots \int_{-\infty}^{\infty} dv_1 \\
&\times \exp\left(-\frac{|v|^2 + r^2}{2}\right) \log\left(1 + \sum_{k' \neq k} \frac{\pi_{k'}}{\pi_k} \exp\left(\frac{|v|^2 - |v - u_{k',k}|^2}{2}\right)\right) r^{m-K} \prod_{i=K}^{m-1} (\sin\theta_i)^{m-i-1}.
\end{aligned}
\tag{B.4}
$$

By the equality

$$\frac{1}{\sqrt{(2\pi)^m}} \int_0^{2\pi} d\theta_{m-1} \int_0^{\pi} d\theta_{m-2} \cdots \int_0^{\pi} d\theta_K \int_0^{\infty} dr \ \exp\left(-\frac{r^2}{2}\right) r^{m-K} \prod_{i=K}^{m-1} (\sin\theta_i)^{m-i-1} = \frac{1}{(2\pi)^{\frac{K-1}{2}}},$$

we conclude (B.2) from (B.4). $\qquad\square$

## B.2 Proof of Theorem 4.6

**Theorem B.2** (Theorem 4.6). *Let $m \geq K \geq 2$. Then, for $s \in (0,1)$,*

$$\left|H[q] - \widetilde{H}[q]\right| \leq \frac{2}{(1-s)^{\frac{K-1}{4}}} \sum_{k=1}^{K} \sum_{k' \neq k} \sqrt{\pi_k \pi_{k'}} \exp\left(-\frac{s\alpha_{k',k}^2}{4}\right).$$

*Proof.* The proof is done by proceeding the argument in proof of Theorem A.2 with the explicit form (B.2) by $|u_{k,k'}| = \|\mu_k - \mu_{k'}\|_\Sigma$. $\qquad\square$

**Theorem B.3.** *Let $m \geq K \geq 2$. Then,*

$$\frac{1}{2} \sum_{k=1}^{K} \sum_{k' \neq k} \frac{\pi_k \pi_{k'}}{1 - \pi_k} \log\left(1 + \frac{1 - \pi_k}{\pi_k} \exp(-2\alpha_{k,k'}^2)\right) \leq \left|H[q] - \widetilde{H}[q]\right|.$$

*Proof.* The proof is given by applying Theorem A.5 in the case of $\Sigma_k = \Sigma$ for all $k \in [K]$ by remarking $c_{k,k'} = 1/2$ in that case. $\qquad\square$

### B.3 Proof of Corollary 4.7

**Corollary B.4** (Corollary 4.7). *Let $m \geq K \geq 2$. Assume $\{\mu_k\}_k$ and $\Sigma$ such that*

$$\frac{\Sigma^{-\frac{1}{2}}(\mu_k - \mu_{k'})}{2} \sim \mathcal{N}(0, c^2 I),$$

*for all pairs $k, k' \in [K]$ $(k \neq k')$. Then, for $\varepsilon > 0$ and $s \in (0,1)$*

$$P\left(\left|H[q] - \widetilde{H}[q]\right| \geq \varepsilon\right) \leq \frac{2(K-1)}{\varepsilon(1-s)^{\frac{K-1}{4}}}\left(1 + \frac{sc^2}{2}\right)^{-\frac{m}{2}}.$$

*Proof.* In the same way as in the proof of Corollary A.7, we use the Markov's inequality and the moment generating function for $\chi^2$-distribution to obtain

$$P\left(\left|H[q] - \widetilde{H}[q]\right| \geq \varepsilon\right) \leq \frac{2}{\varepsilon(1-s)^{\frac{K-1}{4}}} \sum_{k=1}^{K} \sum_{k' \neq k} \sqrt{\pi_k \pi_{k'}} E\left[\exp\left(-\frac{sc^2}{4}\frac{\alpha_{k,k'}^2}{c^2}\right)\right]$$

$$\leq \frac{2}{\varepsilon(1-s)^{\frac{K-1}{4}}} \underbrace{\sum_{k=1}^{K} \sum_{k' \neq k} \sqrt{\pi_k \pi_{k'}}}_{\leq \sum_{k=1}^{K} \sum_{k' \neq k} \frac{\pi_k + \pi_{k'}}{2} = K-1} \left(1 - 2\left(-\frac{sc^2}{4}\right)\right)^{-\frac{m}{2}}$$

$$\leq \frac{2(K-1)}{\varepsilon(1-s)^{\frac{K-1}{4}}}\left(1 + \frac{sc^2}{2}\right)^{-\frac{m}{2}},$$

where we used the upper bound in Theorem B.2 in the first inequality. $\square$

## C  Details of experiments

### C.1  Details of Section 5.1

We give a detailed explanation for the relative error experiment in Section 5.1. We restricted the setting of the experiment to the case for the coincident covariance matrices equation B.1 and minimal number of the mixture components $K = 2$. Furthermore, we assumed that $\Sigma = I$, $\mu_1 = 0$, and $\mu_2 \sim \mathcal{N}(0, (2c)^2 I)$. In this setting, we varied the dimension $m$ of Gaussian distributions from 1 to 500 for certain parameters $(c, \pi_k)$, where we sampled $\mu_2$ 10 times for each dimension $m$. As formulas for the entropy approximation, we employed $\widetilde{H}_{\mathrm{ours}}[q_\theta]$, $\widetilde{H}_{\mathrm{Huber}(R)}[q_\theta]$, and $\widetilde{H}_{\mathrm{Bonilla}}[q_\theta]$ for the method of ours, Huber et al. (2008), and Bonilla et al. (2019), respectively, as follows:

$$\widetilde{H}_{\mathrm{ours}}[q_\theta] = \frac{m}{2} + \frac{m}{2}\log 2\pi + \frac{1}{2}\sum_{k=1}^{K}\pi_k \log |\Sigma_k| - \sum_{k=1}^{K}\pi_k \log \pi_k, \tag{C.1}$$

$$\widetilde{H}_{\mathrm{Huber}(R)}[q_\theta] = -\sum_{k=1}^{K}\pi_k \int \mathcal{N}(w|\mu_k, \Sigma_k)$$

$$\times \sum_{i=0}^{R}\frac{1}{i!}\left((w - \mu_k) \odot \nabla_{\widetilde{w}}\right)^i \log\left(\sum_{k'=1}^{K}\pi_{k'}\mathcal{N}(\widetilde{w}|\mu_{k'}, \Sigma_{k'})\right)\Bigg|_{\widetilde{w}=\mu_k} dw, \tag{C.2}$$

$$\widetilde{H}_{\mathrm{Bonilla}}[q_\theta] = -\sum_{k=1}^{K}\pi_k \log\left(\sum_{k'=1}^{K}\mathcal{N}(\mu_k|\mu_{k'}, \Sigma_k + \Sigma_{k'})\right), \tag{C.3}$$

where equation C.1 is the same as described in Section 3, equation C.2 is based on the Taylor expansion (Huber et al., 2008, (4)), and equation C.3 is based on the lower bound analysis (Bonilla et al., 2019, (14)).

In the case for the coincident and diagonal covariance matrices $\Sigma_k = \text{diag}(\sigma_1^2, \ldots, \sigma_m^2)$, equation C.2 for $R = 0$ or 2 can be analytically computed as

$$\widetilde{H}_{\text{Huber}(0)}[q_\theta] = -\sum_{k=1}^{K} \pi_k \log \left( \sum_{k'=1}^{K} \pi_{k'} \mathcal{N}(\mu_k | \mu_{k'}, \Sigma_{k'}) \right),$$

$$\widetilde{H}_{\text{Huber}(2)}[q_\theta] = -\sum_{k=1}^{K} \pi_k \left[ \log \left( \sum_{k'=1}^{K} \pi_{k'} \mathcal{N}(\mu_k | \mu_{k'}, \Sigma_{k'}) \right) + \frac{1}{2} \sum_{i=1}^{m} \sigma_i^2 C_{k,i} \right],$$

where

$$C_{k,i} := \frac{g_{0,k} g_{2,k,i} - g_{1,k,i}^2}{g_{0,k}^2}, \quad g_{0,k} := \sum_{k'=1}^{K} \pi_{k'} \mathcal{N}(\mu_k | \mu_{k'}, \Sigma_{k'}),$$

$$g_{1,k,i} := \sum_{k'=1}^{K} \pi_{k'} \frac{\mu_{k,i} - \mu_{k',i}}{\sigma_i^2} \mathcal{N}(\mu_k | \mu_{k'}, \Sigma_{k'}),$$

$$g_{2,k,i} := \sum_{k'=1}^{K} \pi_{k'} \left[ \left( \frac{\mu_{k,i} - \mu_{k',i}}{\sigma_i^2} \right)^2 - \frac{1}{\sigma_i^2} \right] \mathcal{N}(\mu_k | \mu_{k'}, \Sigma_{k'}).$$

In the following, we show the tractable formula of the true entropy, which is used in the experiment in Section 5.1. In the case for the coincident covariance matrices and $K = 2$, we can reduce the integral in equation B.2 to a one-dimensional Gaussian integral as follows:

$$H[q_\theta] = \widetilde{H}[q_\theta] - \sum_{k \neq k'} \frac{\pi_k}{\sqrt{2\pi}} \int_{\mathbb{R}} \exp \left( -\frac{|v|^2}{2} \right) \log \left( 1 + \frac{\pi_{k'}}{\pi_k} \exp \left( \frac{|v|^2 - |v - u_{k',k}|^2}{2} \right) \right) dv.$$

Furthermore, we can choose the rotation matrix $R_k$ in Proposition B.1 such that

$$u_{k',k} = \left[ R_k \Sigma^{-\frac{1}{2}} (\mu_{k'} - \mu_k) \right]_{1:1} \geq 0,$$

that is,

$$u_{k',k} = |u_{k',k}| = \left| \Sigma^{-\frac{1}{2}} (\mu_{k'} - \mu_k) \right| = 2|a|, \quad a := \frac{\Sigma^{-\frac{1}{2}} (\mu_1 - \mu_2)}{2}.$$

Hence, we have

$$|v|^2 - |v - u_{k',k}|^2 = -u_{k',k}^2 + 2v u_{k',k} = -4|a|^2 + 4v|a|.$$

Therefore, by making the change of variables as $v = \sqrt{2}\, t$, we conclude that

$$H[q_\theta] = \widetilde{H}[q_\theta] - \sum_{k=1}^{2} \frac{\pi_k}{\sqrt{2\pi}} \int_{\mathbb{R}} \exp \left( -\frac{|v|^2}{2} \right) \log \left( 1 + \frac{\pi_{k'}}{\pi_k} \exp \left( -2|a|^2 + 2v|a| \right) \right) dv$$

$$= \widetilde{H}[q_\theta] - \sum_{k=1}^{2} \frac{\pi_k}{\sqrt{\pi}} \int_{\mathbb{R}} \exp(-t^2) \log \left( 1 + \frac{\pi_{k'}}{\pi_k} \exp \left( -2|a|^2 + 2\sqrt{2}|a|t \right) \right) dt. \tag{C.4}$$

Note that the integration of (C.4) can be efficiently executed using the Gauss-Hermite quadrature.

## C.2 Details of Section 5.2

We give a detailed explanation for the toy 1D regression experiment in Section 5.2. The task is to learn a curve $y = x \sin(x)$ from a train dataset that consists of 20 points sampled from the noised curve $y = x \sin(x) + \varepsilon$, $\varepsilon \sim \mathcal{N}(0, 0.1^2)$, see Figure 4. To obtain the regression model of the curve, we used the neural network model

as the base model that had two hidden layers and 8 hidden units in each layer with erf activation. Regarding the Bayes inference for the BNN-GM, we modeled the prior as $\mathcal{N}(0, \sigma_w)$ and the variational family as the Gaussian mixture. Furthermore, we chose the likelihood function as the Gaussian distribution:

$$p(y|f(x; w)) = \mathcal{N}(f(x; w), \sigma_y).$$

Then, we performed the SGVB method based on the proposed ELBO (3.6), where the batch size was equal to the dataset size. Hyperparameters were as follows: epochs $= 100$, learning rate $= 0.05$, $\sigma_w = 10^6$, $\sigma_y = 10^{-2}$.

