# OpenReview forum: "Variational Inference with Gaussian Mixture by Entropy Approximation"
_TMLR — Withdrawn by Authors_

### Review · Reviewer_i5Jj · 2024-01-14

**Summary Of Contributions:**

This paper offers a theoretical analysis of the previously proposed approximation of the entropy of a mixture of Gaussians by Gal and Ghahramani, 2016.
(I.e., approximating the entropy with the sum of individual entropy of the mixture components, weighted by the mixture coefficients.)
The authors show that this approximate entropy can be upper and lower bounded based on a distance measure between two Gaussians expressed as a ratio between the distance of the means to the sum of their variances (in the special case of unimodal Gaussians).
They show that the approximation error is negligible when this distance is large enough.

**Audience:**

Yes

**Claims And Evidence:**

Yes

**Requested Changes:**

As I mentioned before, the main issue of this paper is the practical consequence of the analysis. It would greatly enhance this paper if the authors could demonstrate some practical applications. Note that, it doesn't have to be in BNNs. In this case, it would be better if the paper is rewritten so that it is independent of BNNs, and thus be more general.

**Strengths And Weaknesses:**

The strength of this work is in the analysis of the approximation of the entropy of a mixture of Gaussians (MoGs).
Specifically, the explicit, easy-to-compute, and intuitive upper and lower bounds of the approximation error are useful.
The proposed distance measure $\alpha$ might also be useful for other analyses of Gaussian-based BNNs (not just variational inference).

The weakness of this paper is the practical implication of the insights.
I find it hard to imagine any practical implication of knowing the approximation quality of the entropy of the mixture Gaussians in the ELBO of variational BNNs.
Indeed, variational BNNs (not just with MoGs) themselves seem to have largely fallen out of favor in practice. E.g. deep ensemble or Laplace (with MoGs [1]) or even posterior refinement [2, 3] seem to be better options if one wants multimodal/expressive posteriors in practice.
So, I think variational BNNs might not be a good showcase for the analysis (which I found interesting).

Also, since the authors proposed BNN-GM as a method, one way to alleviate this weakness is to show that BNN-GM is better than e.g. Laplace-MoGs, which seems to be the state of the arts in terms of MoG-based BNNs. Alternatively, it would be interesting to expand the setup so that BNN-GM accepts MoGs as a prior (currently, BNN-GM only accepts unimodal Gaussian prior)---then the authors can use BNN-GM to refine Laplace-MoGs further.


**References**

1. https://arxiv.org/abs/2111.03577
2. https://arxiv.org/abs/2306.07158
3. https://arxiv.org/abs/2205.10041

---

### Review · Reviewer_tZAu · 2024-01-24

**Summary Of Contributions:**

The reviewed work is concerned with the problem of performing variation inference using Gaussian mixture models. Here, a key challenge is evaluating the entropy of a Gaussian mixture, for which no closed-form expression exists. The reviewed work revisits an approximation introduced by Gal and Gahramani that approximates the entropy of the mixture with a weighted sum of the entropies of its components.
While Gal and Gahramani provide an argument for the convergence of the approximation to the true entropy in a certain limit, they do not provide explicit bounds on the approximation error.

The reviewed work fills this gap by providing upper and lower bounds on the approximation error (and its gradients) as a function of the degree of overlap between the mixture components. The authors further provide a series of numerical experiments comparing different entropy approximations.

The authors also list the approximation formula itself as a contribution but I am unsure what the nature of this contribution is compared to the work of Gal and Gahramani

**Audience:**

Yes

**Broader Impact Concerns:**

No concerns

**Claims And Evidence:**

Yes

**Requested Changes:**

Could you please explain why you list the approximation formula as a contribution in your article?

> We provide the approximation formula for the entropy of the Gaussian mixture in a closed form,
which is suitable for the variational inference (Section 3).

How does it differ from the formula by Gal and Gahramani?

**Strengths And Weaknesses:**

Strength: The reviewed work improves the understanding of popular approximations for the evidence lower bounds.

Weakness: Overall, the paper appears somewhat incremental

---

### Review · Reviewer_zhf2 · 2024-01-24

**Summary Of Contributions:**

- The authors explored the application of the entropy approximation method suggested by Gal and Ghahramani (2016) to estimate the entropy in Variational Inference (VI) using the Gaussian mixture variational distribution.
- Additionally, they established the bounds for the absolute approximation error and its derivatives.
- Through empirical evidence, they demonstrated the effectiveness of Gal and Ghahramani's (2016) approximation method, particularly as the model dimensions increased.

**Audience:**

No

**Broader Impact Concerns:**

I believe that this work does not raise any ethical concerns because it is a theoretical study focused on VI with Gaussian mixture variational distribution in the machine learning context.

**Claims And Evidence:**

No

**Requested Changes:**

To rectify the earlier identified shortcomings, I deem it essential to provide clarifications and corrections to the summarized elements below. If there are any misinterpretations on my part, I would value your guidance in pinpointing them.

- I would appreciate your thoughts on whether this paper should indeed be submitted to a machine learning journal. While the final decision will be made through discussions with the AE, I also feel that submitting to a journal within the field of information science, particularly one with a focus on entropy, might be a viable option.
- If positioning this paper as a machine learning contribution, I would appreciate addressing the following two points:
  - (i) Please consider presenting experimental evidence by utilizing benchmark data for several tasks, including assessing predictive performance and uncertainty calibration, since providing theoretical evidence of the practicality of VI using the proposed method might be challenging.
  - (ii) The paper asserts that a drawback of the proposed method is its computational burden. To strengthen its position as a machine learning paper, please provide a methodology for addressing this issue. While the current technical contribution to VI is considered weak, resolving this problem could enhance the paper's significance. Moreover, addressing this may open avenues for additional contributions, such as further approximation error analyses for new proposed methods.
- In each result of theoretical analysis, please explicitly state why the analysis is beneficial for VI and highlight its importance. Please clarify the motivation behind each analysis and articulate the insights gained from the results in terms of VI.
- I recommend thoroughly reviewing the entire English text to ensure clarity and coherence.

**Strengths And Weaknesses:**

# Strengths And Weaknesses
First and foremost, I would like to express my sincere respect for all the efforts the authors have invested in this paper.
Furthermore, it should be noted that the following review is explicitly written with a clear understanding that TMLR places importance on *accuracy, convincing, and clear evidence,* as well as *capturing readers' interest* over novelty and impact.

## Strengths
- Theoretical analyses were provided on the approximation error of the entropy approximation method originally proposed by Gal and Ghahramani (2016). This not only demonstrates the ability to achieve accurate estimation in VI with Gaussian mixture variational distribution for high-dimensional models, a focal point of this paper, but also ensures the validity of the Bayesian representation of Dropout in deep learning addressed by Gal and Ghahramani (2016).

- In the context of VI utilizing a Gaussian mixture variational distribution, it was shown that Gal and Ghahramani's (2016) approach is effective as an approximation method. From a practical standpoint, it is particularly intriguing in the sense of demonstrating efficacy for high-dimensional models concerning the precision of entropy estimation.

## Weakness
- Concerns regarding the contribution of this paper to the field of machine learning:
  - Firstly, I would like to express concerns regarding whether the contributions of this paper are appropriate for a machine learning publication and whether it should be featured in TMLR. I acknowledge that this is my personal opinion, and the final decision rests with the editor. In my view, the substantive contribution of this paper is primarily confined to the error analysis of entropy approximation. Therefore, I believe it might be more fitting for a journal that predominantly focuses on information theory or entropy, rather than being categorized as a machine learning paper.
  - The entropy approximation method proposed in Sec.3 is essentially much the same as Gal and Ghahramani (2016), with the modification being the adaptation of the method to a specific case of VI. Consequently, asserting this as a contribution seems somewhat weak.
  - Therefore, the main contribution of this paper can be considered as the analysis of the approximation error when applying their entropy approximation method to a Gaussian mixture distribution. In this case, their contribution seems more geared towards **providing an understanding of the mathematical properties of entropy approximation methods** rather than being specifically relevant to the machine learning context. Hence, careful consideration is needed to determine whether categorizing this paper as a machine learning journal paper is appropriate.
  - Given the relatively weak contribution in terms of proposing specific algorithms, positioning this paper as a machine learning contribution would require, at the very least, numerical experiments based on standard benchmarks (such as UCI benchmark data, along with MNIST or CIFAR-10) using the approach in Sec.3 to demonstrate the effectiveness (or ineffectiveness thereof) of VI. Additionally, proposing strategies to overcome the computational burden mentioned in Sec.6, as well as addressing the drawbacks they have identified, could lend more credibility to the claim of being a machine learning paper. Furthermore, explaining the theoretical implications of the results presented in Sec.4, such as the analysis of the gradient of the approximation error provided in Thm4.4, is essential.
  - As it stands, this paper is primarily theoretical in nature focusing on the entropy approximation, machine learning is utilized only for simulations to verify theoretical results. In my perspective, it does not currently align with the category of machine learning papers, especially those related to VI.
  - For VI experiments, referencing works like [Blundell et al., 2015; Pérez-Ortiz et al., 2021], which use algorithms based on reparameterization such as Bayes-by-backprop, could be insightful. Since these implementations are available on Github, adapting the proposed method by changing the variational distribution to a Gaussian mixture might provide the necessary experimental results for comparison.

- Concerns regarding the quality of the presentation:
  - I understand the challenges of conveying contributions effectively in a presentation, especially when English is not the first language. However, currently, there are noticeable grammar mistakes and typos that could have been prevented with a pre-submission review. These issues significantly impact readability and require improvement to capture the interest and engagement of TMLR readers. For instance:
  - Sec.1: "Hence, we approximate... as the sum of the entropy of the unimodal Gaussian equation 3.5" --> "the unimodal Gaussian "expressed in" equation 3.5"?
  - Sec.1: "However, the order and number are hyperparameters..." --> "However, these (= the order of Taylor approximation and the number of splitting) are hyperparameters"? If no, The number of what?
  - Sec.1: "However, our method has a canonical statistical nature as Bayesian model, which is absent in the deep ensemble." --> I apologize, but I am having difficulty understanding the content. I would appreciate it if the authors could clarify this point.
  - Sec.3, equation between (3.1) and (3.2): It would be better to modify from "," to "." since this sentence seems to be completed before the next sentence ("the second...").
  - Sec.5.2: "From the result in Figure 3, we can observe the following." --> "... the following "facts"?

## Concerns regarding the interest of TMLR readers
- Given that TMLR is a journal in the field of machine learning, readers would likely expect technical and theoretical contributions directly relevant to VI when encountering this paper. However, the current state of the paper mainly suggests that the proposed entropy approximation error becomes more precise in high dimensions. Although they hint at potential applicability in the context of Bayesian Neural Networks (BNNs), there is limited indication of how practically and effectively the proposed inference based on this approximation method performs for VI with a variational distribution composed of a Gaussian mixture. In this regard, there is a concern about whether the paper adequately captures the interest of TMLR's assumed readership and addresses the practical effectiveness of the proposed method.
- It is speculated that the insights analyzed in Sec.4 may be more appreciated in a journal where readers are seeking knowledge related to entropy estimation rather than in a general machine learning journal.
- Therefore, it might be beneficial to emphasize the relevance and applicability of the proposed method to VI with Gaussian mixture variational distributions in practical machine learning scenarios (e.g., showing the performance of the proposed method under the more realistic experimental settings), aligning more closely with TMLR's focus and readership expectations.

**Citation**:

[Blundell et al., 2015] C. Blundell et al. Weight Uncertainty in Neural Networks. ICML2015.
https://arxiv.org/pdf/1505.05424.pdf

[Pérez-Ortiz et al., 2021] M. Pérez-Ortiz et al. Tighter Risk Certificates for Neural Networks. JMLR.
https://jmlr.csail.mit.edu/papers/volume22/20-879/20-879.pdf

---

### Note · Authors · 2024-01-28

**Comment:**

Dear Reviwers and Editors,

Thank you very much for handling our paper.
I understand that our paper needs careful considerations before submittion.
So, I would like to withdrawal our submittion to TMLR, and I will revise our papers
based on variable comments we got here.

Thank you and best regrad.

**Withdrawal Confirmation:**

I have read and agree with the venue's withdrawal policy on behalf of myself and my co-authors.